# Different response of surface temperature and air temperature to deforestation in climate models

Johannes Winckler[1,2,3], Christian H. Reick[1], Sebastiaan Luyssaert[4], Alessandro Cescatti[5], Paul C. Stoy[6], Quentin Lejeune[7,8], Thomas Raddatz[1], Andreas Chlond[1], Marvin Heidkamp[1,2], and Julia Pongratz[1,9]

[1]Max Planck Institute for Meteorology, Hamburg, Germany.
[2]International Max Planck Research School on Earth System Modeling, Hamburg, Germany.
[3]Current affiliation: Laboratoire des Sciences du Climat et de l'Environnement, LSCE/IPSL, CEA-CNRS-UVSQ, Université Paris-Saclay, Gif-sur-Yvette, France.
[4]Vrije Universiteit Amsterdam, Faculty of Science, Amsterdam, the Netherlands.
[5]European Commission, Joint Research Centre, Institute for Environment and Sustainability, Ispra, Italy.
[6]Department of Land Resources and Environmental Sciences, Montana State University, Bozeman, MT, USA.
[7]Institute for Atmospheric and Climate Science, ETH-Zürich, Switzerland.
[8]Current affiliation: Climate Analytics, Berlin, Germany.
[9]Ludwig-Maximilians-Universität München, Munich, Germany.

**Correspondence:** Johannes Winckler(johannes.winckler@lsce.ipsl.fr)

**Abstract.** When quantifying temperature changes induced by deforestation (e.g. cooling in high latitudes, warming in low latitudes), satellite data, in-situ observations and climate models differ concerning the height at which the temperature is typically measured/simulated. In this study the effects of deforestation on surface temperature, near-surface air temperature and lower atmospheric temperature are compared by analyzing the biogeophysical temperature effects of large-scale deforestation in the Max-Planck-Institute Earth System Model (MPI-ESM) separately for local effects (which are only apparent at the location of deforestation) and nonlocal effects (which are also apparent elsewhere). While the nonlocal effects (cooling in most regions) influence the temperature of the surface and lowest atmospheric layer equally, the local effects (warming in the tropics but a cooling in the higher latitudes) mainly affect the temperature of the surface. In agreement with observation-based studies, the local effects on surface and near-surface air temperature respond differently in the MPI-ESM, both concerning the magnitude of local temperature changes and the latitude at which the local deforestation effects turn from a cooling to a warming (at 45-55° N for surface temperature and around 35° N for near-surface air temperature). Subsequently, our single-model results are compared to model data from multiple climate models from CMIP5. This inter-model comparison shows that in the northern mid latitudes, both concerning the summer warming and winter cooling, near-surface air temperature is affected by the local effects only about half as strongly as surface temperature. This study shows that the choice of temperature variable has a considerable effect on the observed and simulated temperature change. Studies about the biogeophysical effects of deforestation must carefully choose which temperature to consider.

## 1 Introduction

Afforestation has been proposed as a tool to mitigate climate change globally (UNFCCC, 2011), mainly because forests can store large amounts of carbon (Luyssaert et al., 2008; Le Quéré et al., 2017). In addition, changes in forest cover can cause a warming or cooling via an alteration of the exchange of energy and water between the Earth's surface and the atmosphere, i.e.

the so-called biogeophysical effects (Bonan, 2008). Earth System models have been employed to assess how these biogeophysical effects affect the temperature of the *surface* (e.g., Bala et al., 2007; Pongratz et al., 2010; Davin and de Noblet-Ducoudré, 2010; Boisier et al., 2012; Devaraju et al., 2015; Li et al., 2016) and the temperature of the *near-surface air* (usually air temperature 2 m above zero-plane displacement height) (e.g., Claussen et al., 2001; Gibbard et al., 2005; Findell et al., 2006; Pitman et al., 2009; Bathiany et al., 2010; de Noblet-Ducoudré et al., 2012; Jones et al., 2013; Luyssaert et al., 2018). The

different temperature variables that are considered in studies about deforestation effects are relevant for different questions and applications. Satellite-based studies on changes in radiometric surface temperature provide important information about the biophysical mechanisms of surface energy partitioning and thereby surface-atmosphere interactions (Duveiller et al., 2018). Compared to changes in surface temperature, changes in air temperature may be considered more relevant for human living conditions because of their importance e.g. for the perceived temperature (e.g., Staiger et al., 2011). Within- and below-canopy

air temperature (which is not included in this study) is the most relevant variable for many organisms that live within forests (e.g., De Frenne et al., 2013, 2019). The coupling between ground temperature and air temperature is strongly influenced by the type of vegetation that covers the surface (Baldocchi, 2013; Melo-Aguilar et al., 2018), but it remains unclear whether surface temperature and near-surface air temperature respond differently to deforestation in climate models. This is the focus of the present study.

An answer to this question could help to reconcile apparent inconsistencies in observation-based studies on the effects of deforestation on surface temperature and air temperature. Studies based on satellite observations investigated changes in radiometric surface temperature resulting from deforestation (Li et al., 2015; Alkama and Cescatti, 2016; Duveiller et al., 2018). These studies reported that deforestation results in a local cooling in the boreal regions (north of approx. 45-55°N) and a warming in lower latitudes, highlighting that the contributions from changes in surface albedo and other surface properties

vary with latitude (Bright et al., 2017). Studies based on observations of air temperature from weather stations and Fluxnet towers (Lee et al., 2011; Zhang et al., 2014) also reported a deforestation-induced boreal local cooling and a warming for lower latitudes, but they indicated that the transition between cooling and warming is located further south (at approx. 35°N). It remains unclear whether part of this apparent inconsistency can be attributed to the different heights above the surface at which temperature changes are considered. In contrast to observations, climate models allow to assess the biogeophysical effects

both on surface temperature, on near-surface air temperature, and temperature of the atmosphere within a single consistent framework, rendering climate models suitable tools to investigate this question.

Both air and surface temperature are affected by local and nonlocal biogeophysical effects of deforestation. We define local effects as effects that are only apparent in deforested locations and nonlocal effects as effects that are also apparent in non-deforested locations (see Sect. 2.1 and Winckler et al., 2017). Local effects can for example be caused by a redistribution of

heat between the surface and the atmosphere (e.g., Vanden Broucke et al., 2015) while the nonlocal effects can be caused by advection (Winckler et al., 2018) or by changes in global circulation (Swann et al., 2012; Devaraju et al., 2015; Lague and Swann, 2016). In this study, local and nonlocal effects are analyzed separately for three reasons. First, the difference between local and nonlocal effects matters for decision makers: the local effects may be relevant for policies that aim at adapting to a warming climate locally because they link the climate effects to the areas where policies are implemented (Duveiller et al., 2018). The nonlocal effects are also relevant for international policies that aim at mitigating global climate change because the nonlocal effects may dominate the global mean biogeophysical temperature response to deforestation (Winckler et al., 2018). Second, the observation-based data-sets only record the local effects when comparing temperature between nearby locations with and without forest, or between locations with and without deforestation. The nearby locations share the same background climate, and thus the nonlocal effects cancel out when temperature differences between the locations are considered (Lee et al., 2011; Li et al., 2015; Alkama and Cescatti, 2016; Duveiller et al., 2018). For a consistent comparison to observation-based data-sets, the local effects need to be separated from the nonlocal effects when analyzing climate model results. The third reason to consider local and nonlocal temperature changes separately is that different mechanisms trigger local and nonlocal temperature changes (Winckler et al., 2017). If surface and air temperature respond differently to deforestation, it is unclear whether this difference arises from mechanisms that trigger the local temperature changes (predominantly via changes in the turbulent heat fluxes (Winckler et al., 2018)), mechanisms that trigger the nonlocal temperature changes (predominantly via the incoming radiation that reaches the surface (Winckler et al., 2018)), or from both. A separate analysis of local and nonlocal temperature changes facilitates an investigation of the mechanisms that may cause a different response of surface and air temperature to deforestation.

Here, we investigate how deforestation in the MPI-ESM (Max-Planck-Institute Earth System Model) affects surface and air temperature differently and analyze this separately for the local and nonlocal effects. A previous study contrasted the response of surface temperature only with the response of near-surface air temperature and found mainly differences between surface and air temperature for the local effects (Appendix C in Winckler et al., 2017). We go beyond this previous study by additionally analyzing the effects on temperature in the lowest atmospheric layer and by using simulations with an interactive ocean which enables us to better capture the nonlocal temperature effects of deforestation from the surface to the lower atmosphere (Davin and de Noblet-Ducoudré, 2010). To further analyze the mechanisms that are responsible for differences in these three temperature variables, we investigate the local effects separately for the response in mean daily minimum and maximum temperature. To test the robustness of our results for this particular climate model, we compare the simulation results of the MPI-ESM to existing simulation results from multiple climate models from the Climate Model Intercomparison Project CMIP5. In this inter-model comparison, we contrast the response of the local effects on near-surface air temperature and surface temperature.

## 2 Methods

### 2.1 Simulations of large-scale deforestation in the MPI-ESM

Using the fully coupled climate model MPI-ESM (Giorgetta et al., 2013), the temperature response to deforestation at the surface, at $2\,\mathrm{m}$ and the lowest layer of the atmosphere are obtained from simulations of large-scale deforestation. 550 years of simulations are performed in T63 atmospheric resolution (about $1.9°$) and the last 200 years (which are free of substantial trends in the investigated variables (not shown)) are used for the analysis. Two simulations are performed: a first simulation ('forest world') with forest plant functional types on all areas that can potentially be covered with vegetation (i.e. forests do not exist in deserts etc., Fig. S1). These vegetated areas are taken from a previous study (Pongratz et al., 2008), which derived a map of potential vegetation from remote sensing (Ramankutty and Foley, 1999). The non-forest plant functional types were replaced by the forest types occurring in that grid cell (preserving the relative fraction of the different forest types). In a second simulation forests in the forest world are completely replaced by grasslands in three of four grid boxes in a regular spatial pattern (Fig. S1, equivalent to simulation '3/4' in a previous study (Winckler et al., 2018)). In both simulations, atmospheric $CO_2$ concentrations are prescribed at pre-industrial level in order to obtain only the biogeophysical effects of deforestation. The total, i.e. local plus nonlocal, biogeophysical deforestation effects are then computed as the differences (e.g., in temperature) between these two simulations.

Following the approach in Winckler et al. (2017), the total effects can be separated into the local and nonlocal effects of deforestation as follows:

$$\Delta T_{total} = \Delta T'_{local} + \Delta T_{nonlocal}, \tag{1}$$

where $\Delta T_{total}$ are the temperature changes that are simulated at a deforested grid box and $\Delta T'_{local} = \Delta T_{local} + \Delta T_{local \times nonlocal}$ includes both the local effects and possible interactions between local and nonlocal effects. However, such interactions were found to be small across a wide range of deforestation scenarios (Winckler et al., 2017), and in the following we refer $\Delta T'_{local}$ as 'local effects'. The nonlocal effects are determined from non-deforested grid boxes, where only the nonlocal effects are present. The nonlocal effects are spatially interpolated to the deforested grid boxes using bi-linear interpolation. The local effects at deforested grid boxes can thus be obtained by subtracting the nonlocal effects from the simulated total effects:

$$\Delta T'_{local} = \Delta T_{total} - \Delta T_{nonlocal} \tag{2}$$

The local effects that are obtained this way are similar to the local effects that were obtained in previous studies by comparing temperatures in sub-grid tiles (Malyshev et al., 2015; Schultz et al., 2017; Meier et al., 2018). In contrast to their methods, the method that is applied in this study includes local feedbacks between surface and atmosphere leading e.g. to local changes in clouds or precipitation. In addition, the method used in this study allows us to assess the local effects on the temperature of the lowest layer of the atmosphere (which in most models is not calculated separately for sub-grid tiles) and to assess nonlocal effects on temperature. The choice of deforesting three of four grid boxes is to some extent arbitrary; varying the spatial extent of deforestation influences the magnitude of the nonlocal effects on surface temperature (Winckler et al., 2018), but the local

effects on surface temperature within a grid box are largely insensitive to deforestation elsewhere (Winckler et al., 2017). A detailed description and discussion of the separation approach can be found in Winckler et al. (2017).

## 2.2 Temperature of the surface, the lowest atmospheric layer, and near-surface air in the MPI-ESM

This study investigates the response of three types of temperature to deforestation in the MPI-ESM: surface temperature (T$_{surf}$), temperature of the lowest atmospheric layer (T$_{atm}$), and near-surface air temperature (T$_{2m}$, called 'tas' in CMIP5). Although these temperature variables are part of the standard output of climate model simulations, different models may calculate them differently. In the following, details are provided on the calculations of these variables in the MPI-ESM.

The surface temperature T$_{surf}$ in the MPI-ESM is determined by solving the surface energy balance equation in a bulk canopy layer. For simplicity this layer has a heat capacity that is independent of the vegetation type. This bulk canopy layer exchanges heat with deeper soil layers via the ground heat flux.

It is not possible to assign one geometrical height to the surface layer in the MPI-ESM because there is an internal inconsistency between the two different aspects that are involved in the process of solving the surface energy balance equation: the calculation of the surface radiative budget (absorption of solar radiation and emission of terrestrial radiation to the atmosphere) and the calculation of the turbulent heat fluxes (latent and sensible heat). From the perspective of the radiative budget, the surface is where this radiative budget is calculated (i.e. where the energy balance is solved). In the presence of vegetation this is somewhere in the canopy, but geometrically its exact height cannot be specified. From the perspective of turbulent fluxes, the geometrical height $d + z_0$ above the surface is where the wind speed would become zero in the wind profile based on Monin-Obukhov theory (Leclerc and Foken, 2014). Here, $z_0$ denotes the aerodynamic roughness length and $d$ is the zero-plane displacement height. This $d$ takes into account the displacement effect exerted by vegetation (Leclerc and Foken, 2014; Campbell and Norman, 1998). Geometrically the height $d + z_0$ may differ from the height where the radiative budget is calculated.

What does this inconsistency imply for the comparison between T$_{surf}$ in the MPI-ESM and satellite-based products? For comparison with satellite observations only the radiative perspective is relevant – because satellites estimate temperature based on the emissions of terrestrial radiation. That the Monin-Obukhov theory provides a different definition of surface height must be considered as a special approximation to solve the energy balance, but has no consequences for comparison with satellite observations of T$_{surf}$.

The 'atmospheric temperature' T$_{atm}$ is defined here as the temperature of the lowest of the 47 atmospheric layers in the MPI-ESM (Stevens et al., 2013). The thickness of this layer is around 60 m (at 15°C), and the temperature is volume-averaged in this layer. This temperature is used for the calculation of the turbulent heat fluxes and T$_{surf}$.

The near-surface air temperature T$_{2m}$ is estimated in the MPI-ESM as temperature of the air at 2 m above $d + z_0$. Because it is unclear (and irrelevant for the calculations) where within the canopy this $d + z_0$ is, a comparison of T$_{2m}$ between the MPI-ESM and observations is challenging, especially in forests (see Sect. 4). The MPI-ESM does not have a representation of within-canopy air temperature or separate temperatures of the surface and the vegetation canopy.

In the MPI-ESM, following Geleyn (1988), $T_{2m}$ is obtained via a procedure based on Monin-Obukhov similarity theory that uses the values at the surface and the lowest atmospheric layer. This procedure employs dry static energy instead of temperature because dry static energy is a conserved quantity in an adiabatic process.

$$s_{z_{aero}} = c_p T_{z_{aero}} + g z_{aero}, \tag{3}$$

where $c_p$ is the heat capacity of moist air, and $g$ is the gravitational acceleration of the Earth, and $s_{z_{aero}}$ and $T_{z_{aero}}$ are the dry static energy and temperature at the aerodynamic height $z_{aero} = z - (d + z_0)$ where $z$ is the height above the surface.

At $2\,\mathrm{m}$ above $d + z_0$, the dry static energy is then obtained as follows:

$$s_{2m} = s_{surf} + (s_{atm} - s_{surf}) \gamma\left(\frac{2\,\mathrm{m}}{z_0}, R_i\right), \tag{4}$$

where $s_{surf}$ and $s_{atm}$ denote the dry static energy at the surface and the lowest atmospheric layer, and $\gamma$ is a nonlinear function based on Monin-Obukhov similarity theory with values ranging between 0 and 1 that depends on the roughness length $z_0$ and on the bulk Richardson number $R_i$, which is closely related to the temperature gradient $T_{surf} - T_{atm}$. Different functions for $\gamma$ are used for near-surface neutral ($R_i \approx 0$), stable ($R_i < 0$), and unstable conditions ($R_i > 0$) (Geleyn, 1988; ECMWF Research Department, 1991, section 3.1.3). Note that both $s_{surf}$ and $s_{atm}$, but also $R_i$ and $z_0$ are affected by deforestation. After this procedure, $T_{2m}$ is obtained as $T_{aero}$ from equation (3) by setting $z_{aero} = 2\mathrm{m}$.

## 2.3 Isolation of local effects across CMIP5 models

The response of $T_{surf}$ and $T_{2m}$ to deforestation in the northern-hemisphere mid latitudes are compared across a wide range of climate models from CMIP5: CanESM2, CCSM4, CESM1-CAMS, GFDL-CM3, HadGEM2-ES, IPSL-CM5A-LR, MPI-ESM-LR, and NorESM1-M. Given that the these models did not simulate the '3/4' deforestation (see Sect. 2.1), we invoke the difference between 'historical' and 'piControl' simulations to isolate the local temperature response from the CMIP5 ensemble (Taylor et al., 2012). The 'historical' simulations are subject to all forcings including changes in greenhouse gases and land use while the 'piControl' simulations are subject to constant boundary conditions and no forcings. To isolate the local effects of deforestation, we use a method that was already applied and validated on these simulations (Lejeune et al., 2018). This method assumes that temperature in neighboring grid boxes can be affected differently by the local effects of deforestation, depending on the forest cover change in each grid box, whereas other climate forcings (like greenhouse gases, but also the nonlocal effects) influence neighboring grid boxes in a similar way. Linear regressions are fitted between temporal changes in temperature (the so-called predictand) and forest cover change (the so-called predictor) within a spatially moving window encompassing 5 x 5 model grid boxes. In the center of this moving window, the local effects are then defined as the temperature change for a hypothetical conversion of 100% forest into 100% open land (given by the slope of the regression) and is by construction largely independent of the changes due to the nonlocal greenhouse gas forcing and nonlocal deforestation effects (given by the $y$-intercept of the regression).

We consider here the difference between the last 30 years (1971-2000) of 'historical' simulations for which data in all models are available and 30 years of the pre-industrial control simulations (piControl), from which the temporal changes

in both temperature variables and forest fraction since 1860 are computed. The 'historical' simulations consist of several ensemble members for each model, where each ensemble member experiences the same forcings but starts from different initial conditions. The moving-window method is applied to several combinations of ensemble members from the 'historical' simulations and time slices from the 'piControl' simulations for each model, and the number of analyzed ensemble members is shown in Table S1.

For this inter-model comparison, we focus on the local effects for two reasons: (1) The nonlocal deforestation effects cannot be isolated from the analyzed set of simulations because the nonlocal deforestation effects cannot be distinguished from nonlocal greenhouse gas forcing in these simulations. (2) The local effects exhibit a better signal/noise ratio compared to the nonlocal effects (e.g., Lejeune et al., 2017). This is important because the climate variability can be large compared to the nonlocal effects for the short time spans (30 years) that are analyzed here (Winckler et al., 2018). Furthermore, climate variability is especially large in the mid-latitudes (Deser et al., 2012) that are analyzed in the inter-model comparison.

## 3  Results

### 3.1  Different temperature response of surface and air temperature in the MPI-ESM

In the MPI-ESM, deforestation in three of four grid boxes triggers substantial nonlocal cooling in most regions (Fig. 1). This happens because deforestation locally reduces the input of latent and sensible heat from the surface to the atmosphere (Winckler et al., 2018). Thus, the atmosphere becomes cooler and drier (not shown), and this leads to a reduction of $T_{surf}$ in many regions, mainly because of reduced longwave incoming radiation (Davin and de Noblet-Ducoudré, 2010; Winckler et al., 2018). The spatial pattern of these nonlocal effects is very similar for $T_{surf}$, $T_{2m}$ and $T_{atm}$.

In contrast to the nonlocal effects, the local effects differ strongly between $T_{surf}$, $T_{2m}$, and $T_{atm}$. Deforestation strongly influences the local surface energy balance: the imposed changes in surface properties in the model (surface albedo, evapotranspirative efficiency and surface roughness) cause a surface warming for the local effects in most regions, except for the high northern latitudes where the local effects cause a surface cooling (Fig. 1). The changes in surface properties influence not only the local $T_{surf}$ but also the flux of sensible heat from the surface into the lower boundary layer (not shown). Intuitively one would expect that the change in sensible heat flux alters $T_{atm}$, e.g. an increased input of sensible heat into the atmosphere could raise the temperature of the atmospheric air above a deforested location. However, in our model results $T_{atm}$ is largely unaffected by the local effects of deforestation (Fig. 1). We interpret this lack of local effects in $T_{atm}$ as follows: it takes long enough for the lowest atmospheric layer to warm up (due to the deforestation-induced increase in sensible heat flux) for the heated air to be transported to higher atmospheric layers and the neighboring grid boxes. Due to the advection, the change in $T_{atm}$ is hence not only seen in a deforested location but also in nearby grid boxes that are not deforested. Thus, this warming or cooling is accounted for in the nonlocal effects. In the nearby grid boxes, the change in $T_{atm}$ and/or atmospheric moisture can then influence also $T_{surf}$ via changes in longwave incoming radiation (Davin and de Noblet-Ducoudré, 2010; Winckler

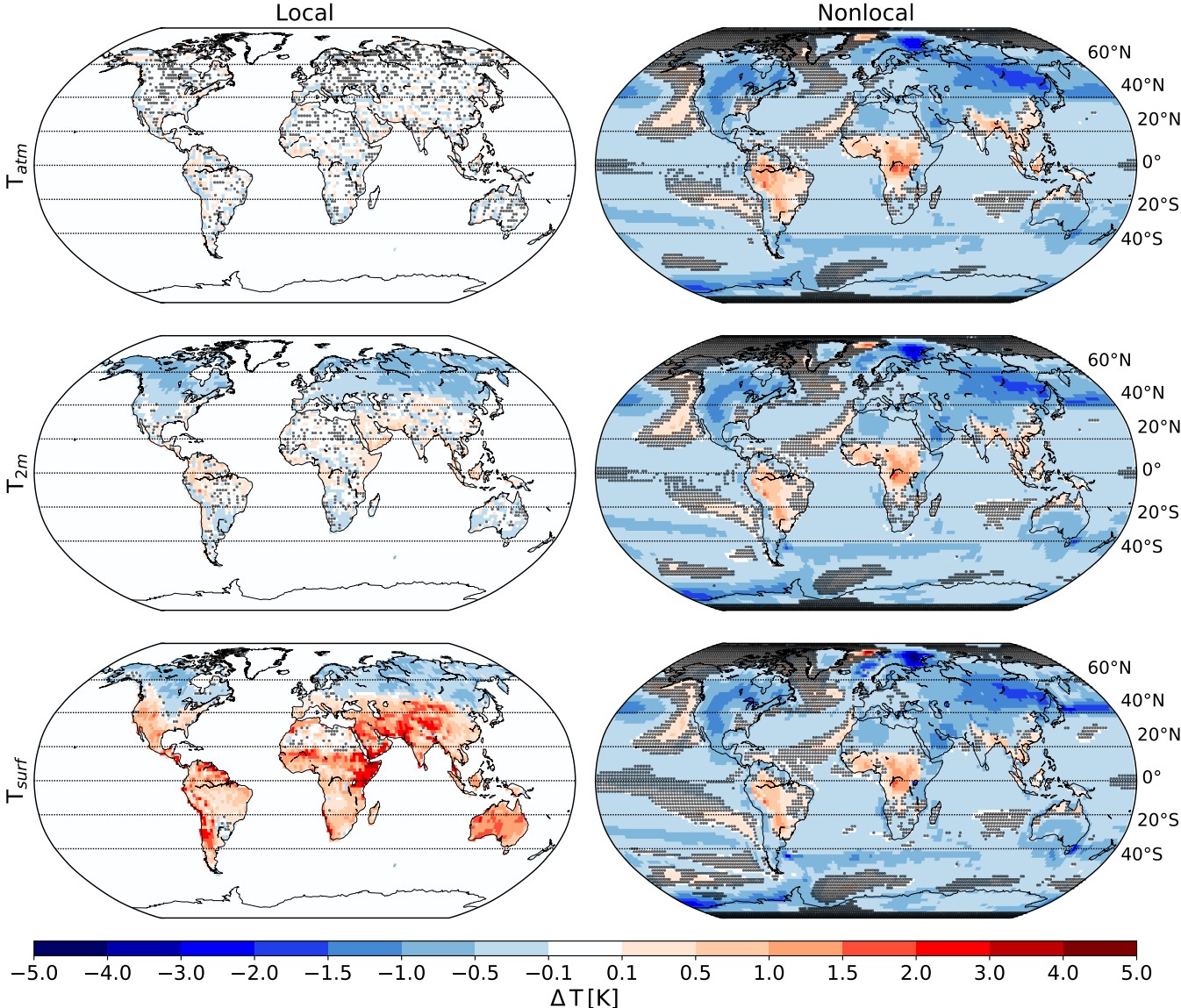

**Figure 1.** Deforestation-induced annual mean response of temperature in the lowest atmospheric layer ($T_{atm}$), near-surface air ($T_{2m}$), and at the surface ($T_{surf}$). Deforestation was applied to three of four grid boxes (Fig. S1). Stippling indicates where results are not statistically significant at a 5% level for a Student's t test accounting for lag-1 autocorrelation (Zwiers and von Storch, 1995). Zonal averages are shown in Fig. S2.

et al., 2018), which could explain why the nonlocal effects are similar for $T_{atm}$ and $T_{surf}$. While in $T_{atm}$, advection can lead to a direct exchange of heat between neighboring grid cells, the same is not possible for $T_{surf}$; there is no direct horizontal

exchange of heat between the surface of neighboring grid cells, and this difference (advection for $T_{atm}$ but not $T_{surf}$) may explain why local effects can be seen in $T_{surf}$ but not in $T_{atm}$.

Because $T_{2m}$ is an interpolation between $T_{surf}$ and $T_{atm}$, we expected that also the local response of $T_{2m}$ would lie in between the response of $T_{surf}$ and $T_{atm}$. In a lot of regions this is the case, but in other regions, most notably those that show a cooling, the local effects on $T_{2m}$ seem to cool more than $T_{surf}$, and in some regions even the sign differs between $\Delta T_{surf}$ and $\Delta T_{2m}$ (e.g. parts of the US and regions in the southern extra-tropics, Fig. S3). The different response of $T_{surf}$ and $T_{2m}$ in relation to the observation-based findings is discussed in Sect. 4.

To better understand the apparent discrepancy between $\Delta T_{surf}$ and $\Delta T_{2m}$, we separately analyze the local temperature response for boreal winter (DJF) and summer (JJA), and the response of mean daily minimum temperature ($T_{min}$, which approximately corresponds to nighttime conditions) and maximum temperature ($T_{max}$, which approximately corresponds to daytime conditions). For $T_{surf}$, the response to deforestation locally differs strongly between DJF, JJA, $T_{min}$, and $T_{max}$ values (Fig. 2). For northern-hemisphere DJF and $T_{min}$, deforestation leads to a local $T_{surf}$ cooling, while for JJA and $T_{max}$, deforestation leads to a local $T_{surf}$ warming. This is qualitatively in good agreement with observation-based studies that show a local cooling in the boreal regions in DJF (Alkama and Cescatti, 2016; Bright et al., 2017; Duveiller et al., 2018) and in agreement with the local increase in the diurnal amplitude due to deforestation (Li et al., 2015; Alkama and Cescatti, 2016; Schultz et al., 2017). Similarly as in the case of long-term mean temperature (Fig. 1), $T_{atm}$ locally shows little response to deforestation, neither for DJF, JJA, $T_{min}$, nor $T_{max}$ (Fig. 2). For both $T_{surf}$ and $T_{2m}$, the annual mean response then depends on the balance between the daytime and nighttime response, and the balance between the responses in different seasons.

For near-surface air temperature, the $T_{max}$ response is substantially weaker, and in many areas of opposite sign, than for the surface, similar to the lowest atmospheric layer (land mean absolute changes for $T_{max}$: 1.62 K for $T_{surf}$, 0.19 K for $T_{2m}$, and 0.10 K for $T_{atm}$). On the contrary, most regions exhibit a strong $T_{min}$ cooling of $T_{2m}$, similar to the $T_{min}$ response of $T_{surf}$ (land mean absolute changes for $T_{min}$: 0.67 K for $T_{surf}$, 0.48 K for $T_{2m}$, and 0.10 K for $T_{atm}$).

$T_{max}$ responds differently for $T_{surf}$ and $T_{2m}$ not only for annual mean $T_{max}$, but in some tropical/subtropical regions also for $T_{max}$ during DJF and JJA (Fig. S9). We interpret this as follows: During daytime when $T_{max}$ occurs, the $T_{surf}$ is higher than the temperature of the lowest atmospheric layer (Fig. S4) because the surface is heated by incoming radiation. In accordance with previous studies (e.g., Li et al., 2015), deforestation further increases $T_{surf}$ (Fig. 2) and thus also the difference between $T_{surf}$ and $T_{atm}$ (illustrated in Figure S5b). Intuitively, one would expect that an increase in this difference would result also in an increase in $T_{2m}$. But by the increased surface temperature, the well mixed zone in the boundary layer not only extends to larger heights, but also extends further down. Accordingly, the cooler air from above mixes further down, and if this affects heights below 2m, $T_{2m}$ is lowered. In the model's calculation of $T_{2m}$, a deforestation-induced increase in $T_{surf}$ increases the difference between $T_{atm}$ and $T_{surf}$ and thus leads to an even more negative Richardson number $R_i$. This Richardson number enters the similarity function for the calculation of $s_{2m}$ ($\gamma$ in equation 4, see underlying report (ECMWF Research Department, 1991)) such that the vertical profile of $s_{2m}$ becomes more non-linear and the calculated $T_{2m}$ gets closer to $T_{atm}$. As a result, daily maximum $T_{2m}$ in the model may decrease although $T_{surf}$ increases.

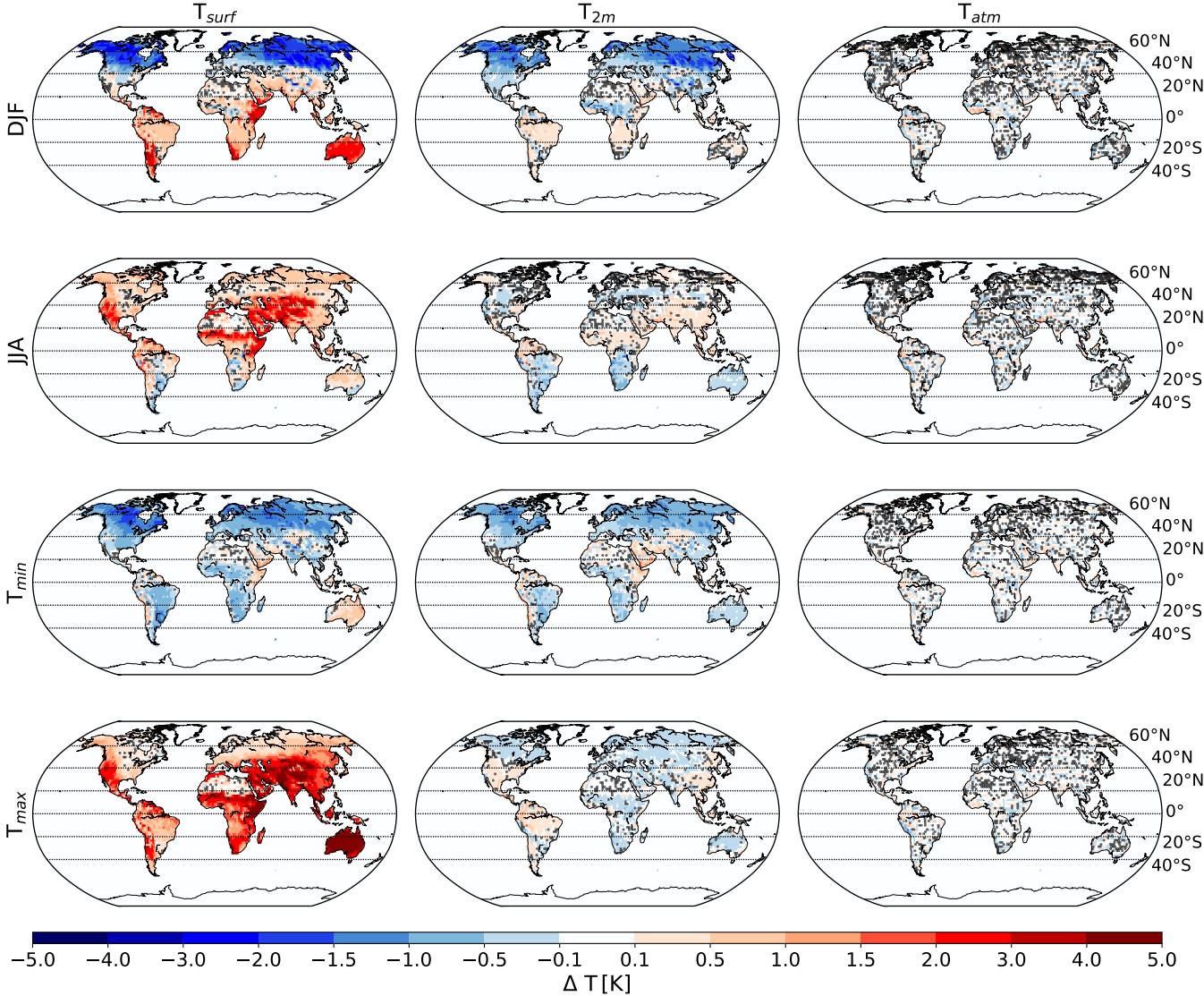

**Figure 2.** Seasonal and diurnal temperature response to the local effects of deforestation, separately for boreal winter (DJF) and summer (JJA), daily maximum ($T_{max}$) and minimum temperature ($T_{min}$). Stippling indicates where results are not statistically significant at a 5% level for a Student's t test accounting for lag-1 autocorrelation (Zwiers and von Storch, 1995).

During nighttime when $T_{min}$ occurs, the surface loses energy via outgoing longwave radiation, and thus the surface is often cooler than the overlying atmosphere (Fig. S4). In accordance with previous studies (e.g., Schultz et al., 2017), deforestation further decreases $T_{surf}$ and thus $T_{surf}$ and $T_{atm}$ further diverge. For $T_{2m}$, one expects that less vertical mixing than during daytime results in $T_{2m}$ tending towards $T_{surf}$. In the model's calculation of $T_{2m}$, this behavior is a result of the $\gamma$ in eq. (4) being less non-linear in stable compared to unstable conditions (for the concrete form of the $\gamma$ function see ECMWF Research

Department (1991)) with the consequence that $T_{2m}$ at deforested locations stays closer to $T_{surf}$ compared to daytime conditions (Figure S5a). As a result, in most regions the calculated $T_{2m}$ follows the deforestation-induced nighttime surface cooling but not necessarily the deforestation-induced daytime surface warming.

The different $T_{2m}$ responses for DJF and JJA in the model may be caused by the same mechanism that may be responsible for the different behaviors for $T_{min}$ and $T_{max}$. In JJA, when the surface is often warmer than the overlying atmosphere, deforestation leads to a further local $T_{surf}$ warming (Fig. 2). Similar to the case of $T_{max}$, $T_{2m}$ barely responds or even responds with a cooling in some regions. In DJF, when the surface is often cooler than the overlying atmosphere, deforestation leads to a local $T_{surf}$ cooling. Similar to the case of $T_{min}$, also $T_{2m}$ shows a substantial cooling in most northern-hemisphere regions. The reason why $T_{2m}$ responds similar to $T_{surf}$ in northern-hemisphere DJF but not JJA may be similar to the reason why $T_{2m}$ responds similar to $T_{surf}$ for $T_{min}$ but not for $T_{max}$, as described in the previous two paragraphs. Our findings are in qualitative agreement with the findings of (Meier et al., 2018) (their Fig. 9), who found a strong deforestation-induced daytime $T_{surf}$ warming and a moderate $T_{2m}$ cooling in many regions using the CLM (Community Land Model). Whether or not $T_{surf}$ and $T_{2m}$ respond similarly to deforestation may strongly depend on how $T_{2m}$ is calculated in the respective climate model. For the summer and winter response in the northern mid-latitudes, the responses of $T_{surf}$ and $T_{2m}$ is compared across climate models from CMIP5 in the following.

## 3.2 Different temperature response of surface temperature and air temperature across climate models

While the above results refer only to the MPI-ESM climate model, quantitatively the local effects on $T_{surf}$ and $T_{2m}$ also differ in other climate models (Fig. 3). We analyze the local effects of historical deforestation and the average over mid-latitude areas (40-60° N) that experienced intense deforestation ($\geq 15\%$) since 1860. We choose the northern mid-latitudes for two reasons: (1) This is where most historical deforestation happened, and regions with intense deforestation are required in the moving-window approach for isolating the local effects across models (see Sect. 2.3), and (2) the mid-latitudes are a suitable test case because there the local effects on $T_{surf}$ have a different sign in the winter (DJF) and summer season (JJA)(Fig. 2).

In the considered areas, in most models (with the exception of CanESM2 and GFDL-CM3) $T_{surf}$ and $T_{2m}$ respond similarly for changes in annual means (Fig. 3 $a$). This is also true for the MPI-ESM – $T_{surf}$ and $T_{2m}$ respond similarly when averaged over the respective regions (mid-latitude areas (40-60° N) that experienced intense deforestation ($\geq 15\%$) since 1860). A difference in response between $T_{surf}$ and $T_{2m}$ becomes apparent also for the mid-latitudes when analyzing seasons (DJF and JJA separately) instead of annual means. Almost all of the tested models show substantial differences between $T_{surf}$ and $T_{2m}$ at seasonal scale (Fig. 3 $b$-$c$).

In JJA (Fig. 3c), all but one model show a surface warming locally, with the $T_{surf}$ responding more strongly than $T_{2m}$ by a factor of around two (Table S1). Only the HadGEM2-ES climate model is an outlier: there, $T_{surf}$ responds to deforestation with a local cooling (-0.13 K), which is not in agreement with observation-based studies (Li et al., 2015; Alkama and Cescatti, 2016; Bright et al., 2017; Duveiller et al., 2018), and in the HadGEM2-ES $T_{2m}$ cools even more strongly (-0.27 K) than $T_{surf}$. This inter-model comparison confirms that the local deforestation responses of $T_{surf}$ and $T_{2m}$ quantitatively differ strongly

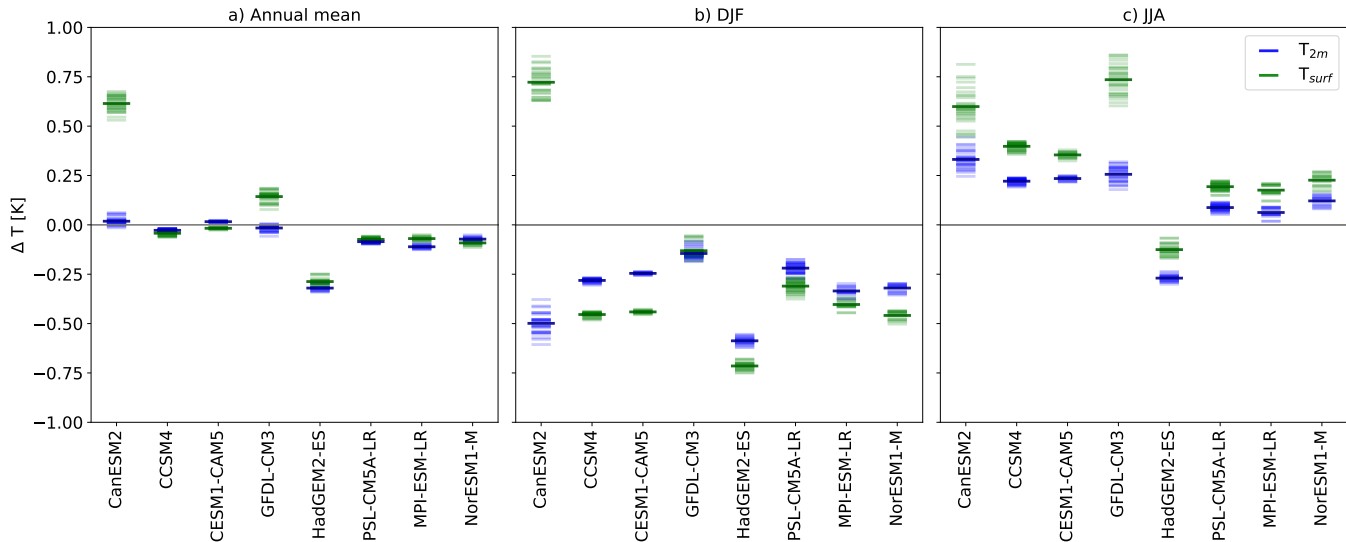

**Figure 3.** Local effects on near-surface air temperature ($T_{2m}$) and surface temperature ($T_{surf}$) for CMIP5 models. Values are averaged over mid-latitude areas (40-60° N) that experienced intense deforestation ($\geq 15\%$) since 1860. Positive values indicate a deforestation-induced warming. Each transparent marker denotes one combination of ensemble members from the 'historical' and 'piControl' experiments, respectively. The solid markers denote the mean values. The corresponding maps are shown in Figs. S6-S8. The local effects are isolated as in the study by Lejeune et al. (2018).

in JJA, but in contrast with the MPI-ESM results shown above (Fig. 2), all models in Fig. 3 show the same sign of the JJA responses for $T_{surf}$ and $T_{2m}$.

 In DJF (Fig. 3b), all but one model show a surface cooling locally, again with $T_{surf}$ responding stronger than $T_{2m}$ in most models. An exception is the CanESM2 model, which locally responds to deforestation with a strong $T_{surf}$ warming and

5  $T_{2m}$ cooling. In some of the other climate models (CCSM4, CESM1-CAM5, NorESM1-M, all sharing the same land surface model CLM4), the $T_{surf}$ cooling is approximately twice the cooling of $T_{2m}$, analogous to the JJA response. In other models (GFDL-CM3, HadGEM2-ES, IPSL-CM5A-LR, MPI-ESM-LR), the two variables respond more similarly in DJF compared to JJA.

 Overall, the inter-model comparison suggests that the quantitatively different response of $T_{surf}$ and $T_{2m}$ is not specific

10  to the MPI-ESM model. In agreement with previous studies (Pitman et al., 2009; Boisier et al., 2012; de Noblet-Ducoudré et al., 2012; Lejeune et al., 2017), the inter-model spread in the temperature response in Fig. 3 is large (e.g., in JJA inter-model (excluding the HadGEM2) standard deviation 0.20 K, inter-model mean 0.38 K). However, the investigated models agree better concerning the ratio between the $T_{2m}$ and $T_{surf}$ response (JJA inter-model (excluding the HadGEM2) standard deviation 0.11, inter-model mean 0.50). Both for DJF and JJA (Table S1), and for most of the investigated models, the ratio of changes in $T_{2m}$

15  and $T_{surf}$ of 0.5 (range 0.35-0.65, excluding HadGEM2) is largely independent of the magnitude and sign of the $T_{surf}$ response. The temperature response of $T_{2m}$ is between zero and the response of $T_{surf}$ with a ratio that depends on the exact

way in which $T_{2m}$ is calculated in the respective models (but most likely all models use a Monin-Obukhov approach). Further studies may investigate to what extent the calculation of $T_{2m}$ differs across models.

## 4   Discussion and conclusions

This study shows that in climate models, surface temperature ($T_{surf}$) and near-surface air temperature ($T_{2m}$) respond differ-
ently to deforestation. In the MPI-ESM, the nonlocal cooling in most regions (present also in locations that were not deforested) of $T_{surf}$ and $T_{2m}$ is similar, while their local warming in the tropics and cooling in the higher latitudes (present only in locations that were deforested) differs. In the northern mid- and high latitudes, the annual mean local cooling of $T_{2m}$ can be stronger than the local cooling of $T_{surf}$, but in most regions $T_{surf}$ responds stronger than $T_{2m}$. Across most models, the local effects of deforestation on $T_{surf}$ and $T_{2m}$ in the mid-latitudes differ by a factor of two, for both local warming during summer
and local cooling during winter.

This study illustrates that the conclusions concerning the effects of deforestation can depend on the considered temperature measure. The differences in magnitude and pattern between $\Delta T_{surf}$ (e.g., Li et al., 2015; Alkama and Cescatti, 2016; Duveiller et al., 2018) and $\Delta T_{2m}$ (e.g., Lee et al., 2011; Zhang et al., 2014) obtained from observation-based studies largely agree with our findings (more cooling for $T_{2m}$ than for $T_{surf}$ in the mid-latitudes for the local effects in the MPI-ESM (see Figs. 1
and S2) and thus our results make it seem plausible that the consideration of different temperature measures can explain some of the discrepancies between the satellite-based and in-situ–based studies. A consistent comparison between satellite–based and in-situ–based studies can be challenging because they may report different variables. Because of the heterogeneous emissivity of the land surface (Jin and Dickinson, 2010), satellite-based data-sets usually report changes in radiometric surface temperature, which represents a combination of temperature at the top of the vegetation and the soil (through gaps in the
canopy). Satellite-based direct estimates of air temperature (based on the intensity of upwelling microwave radiation from atmospheric oxygen) are available only for broad vertical layers of the atmosphere and at coarse spatial scale (Von Engeln and Bühler, 2002). Instead of direct observations, air temperature was derived from surface temperature by empirical methods (Alkama and Cescatti, 2016) or process-oriented models (i.e. by solving the surface energy budget) (Hou et al., 2013). More direct observational investigations on the effects of deforestation on air temperature were based on recordings from weather
stations and Fluxnet towers, which measure temperature at different heights. For instance, weather stations, e.g. in forest clearings, recorded temperatures at a height of between 1.2 and 2.0 m above ground level (WMO, 2008) while Fluxnet sites recorded temperatures typically 2-15 m above forest canopies (Lee et al., 2011; Zhang et al., 2014). The different measurement height may lead to systematic differences because of the steep vertical temperature profile that develops near the surface under stable atmospheric conditions (e.g. at night) especially over open land (Schultz et al., 2017). In contrast to satellite-based
products, which are available at a high spatial resolution, the spatial distribution of Fluxnet towers and weather stations is biased toward developed countries and there is a relatively poor geographical coverage of rural areas in developing countries where

deforestation has occurred recently (Hansen et al., 2013). To perform a meaningful comparison, near-surface air temperature would have to be available at the same height above canopy top (preferably multiple heights) for the various land cover types and with a good geographical coverage.

The comparison of deforestation effects in observations and climate models is even more challenging. First, the respective
variables in the models are only a proxy of the variables that were recorded in observation-based data-sets (see Sect. 2.2 for the MPI-ESM). Second, model-based studies usually analyzed the combination of local and nonlocal effects (especially relevant for simulations of large-scale deforestation where the nonlocal effects can be large), while observation-based studies only analyzed local effects, for which $T_{surf}$ and $T_{2m}$ respond differently (Fig. 4). Any nonlocal effects are excluded from the observations because possible nonlocal effects are present both in forest locations and nearby open land, and thus the nonlocal
effects cancel out when looking at the difference between forests and open land, which is acknowledged by these studies (e.g., Li et al., 2015; Alkama and Cescatti, 2016; Bright et al., 2017; Duveiller et al., 2018). Note that Earth system models consider further climate effects when simulating deforestation-induced releases of land carbon into the atmosphere (e.g., Pongratz et al., 2010; Le Quéré et al., 2017). Because $CO_2$ is a well-mixed greenhouse gas the resulting warming can be expected to act essentially nonlocally and likely influences surface and air temperature similarly.
The different response of surface temperature ($T_{surf}$) and air temperature ($T_{2m}$) is relevant for climate policies. Strategies that aim at adapting locally to warming air temperature may focus on perceived temperature and thus $T_{2m}$, but this study shows that the local effects on $T_{2m}$ may substantially differ from those on $T_{surf}$. Our results for the MPI-ESM suggest that the difference between $T_{2m}$ and $T_{surf}$ is particularly strong for mean daily maximum temperature (see Fig. 2). Further studies may investigate whether this is also true for other climate models and observation-based data-sets. Consequently, strategies
in the agricultural sector that aim at adapting locally to warming soil and canopy temperatures may focus on the local effects on surface temperature because this variable is relevant for the organisms that live there. On the other hand, for international policies that aim at mitigating global warming, what matters is not only temperature at the location of deforestation but also in nearby and in remote regions. Thus, international policies may additionally consider the nonlocal effects. For the nonlocal effects, the response of $T_{surf}$ and $T_{2m}$ are rather similar (Fig. 4) and a distinction between the two temperature measures is
therefore less relevant. To sum up, this study emphasizes that the local biogeophysical effects of deforestation influence $T_{surf}$ and $T_{2m}$ differently, and thus, a careful choice based on the respective application has to be made whether a study should focus on changes in surface temperature or near-surface air temperature.

*Author contributions.* J.W., C.H.R and J.P. designed the research; J.W. performed the simulations with the MPI-ESM, analyzed the data and drafted the manuscript. All authors contributed to the interpretation of the data and revision of the manuscript.

*Competing interests.* The authors declare no competing financial interests.

*Acknowledgements.* Our simulations were performed at the German Climate Computing Center (DKRZ). This work was supported by the German Research Foundation's Emmy Noether Program (PO 1751). PCS acknowledges the U.S. National Science Foundation awards DEB-1552976, and EF-1702029. We want to thank all groups who provided data for the inter-model comparison, and Vivek Arora discussions on the results for the CanESM2. Primary data and scripts used in the analysis and other supporting information that may be useful in reproducing the author's work are archived by the Max Planck Institute for Meteorology and can be obtained by contacting publications@mpimet.mpg.de.

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
