# Peer review of "Different response of surface temperature and air temperature to deforestation in climate models"

_Earth System Dynamics, 2018_

## Referee Comment (RC1) · Anonymous Referee #1 · 6 Nov 2018

**GENERAL COMMENTS**

This study endeavors to tease out the differences in the local response to deforestation on surface temperature and near surface air temperature on global scales as derived from an Earth system model and several climate models from the CMIP5 archive. The study uses a clever approach to first estimate non-local effects by considering only non-deforested grid points and producing a map of non-local effects by interpolation on deforested grid points. The local effect is then the difference between the total signal (total change in temperature due to deforestation) and the non-local effect. The main findings are that 1) deforestation mainly results in a non-local cooling and drying of the lowest atmospheric level, T2m and Tsurf with warming in the tropical land regions, 2) local effects are more strong and heterogeneous at the surface, 3) in the mid-latitudes the local response to deforestation of Tsurf and T2m can be of different magnitudes and sometimes even opposite. Authors then also try to explain this opposite local response of Tsurf and T2m in the mid-latitudes but the reasoning does not come across very clearly and in my opinion should be revised with details. Overall, the study proposes a potential new statistical method (based on the author's previous work) to address some previously observed differences between the response to deforestation of Tsurf and T2m. This is a very important research question pertaining to our understanding of the impacts of deforestation on regional climate. This study points out a very important distinction that should be made while interpreting results from datasets of surface temperature versus near surface air temperature. In this regard the study contributes to current knowledge significantly and so is worthy of consideration. However, several important questions regarding the methodology and physical interpretation of the results remain which need to be addressed. I would like the authors to comment on my questions with some further analysis if possible/needed as seen fit by the authors. My comments are rather minor but I recommend publication of the study after another round of revisions which I'll be happy to review.

**SCIENCE/SPECIFIC COMMENTS**

1. This is probably outside the scope of the present study but one still questions - what is the mechanism that results in opposite responses of Tsurf and T2m in the midlatitudes? Can any mechanism be generalized to all such land regions which show opposite responses of dTsurf and dT2m? Probably not because otherwise all land regions between 35 and 55 north as well as south would show the opposite response. The authors do provide an explanation using the model physics and parametrizations (Page 8, line 29) but it is hard to interpret the underlying physics from this argument. Also it is not clear from this argument why such an opposite response will be observed only in the mid-latitudes. I think it will be worthwhile for the authors to include any
hypotheses about candidate mechanisms in the manuscript? A bit more explanation in the present manuscript is needed if the authors intend to explain this opposite response using the Richardson number, because the argument in its present form is not very clear.

2. The cross product between dTlocal and dTnonlocal have been neglected based on some analysis by previous studies. But there are other non-local factors that can impact and couple with dTlocal, for example precipitation changes due to circulation changes corresponding to a particular pattern of deforestation can bring about changes in Tsurf via the surface energy budget. These changes will be counted as non-local because they are not a direct consequence of local deforestation. So this component of dTsurf should be accounted for in the non-local dTsurf which is estimated using neighboring grind points. But the neighboring grid points could have an entirely different land cover which could result in a nonrepresentative non-local dTsurf at deforested grid points - because the surface energy balance in these grid points will be different due to different vegetation types. So the effect of such a dTsurf can not be obtained from interpolation from neighboring points. How are such non-local effects from changes in variables other than Tsurf, T2m and Tair considered in the methodology? Do the authors think such cross terms will also be negligible as is the case with dTlocal and dTnonlocal? If so can that be explicitly shown?

3. Page 6, line 7- I hope I understand this correctly – so land cover change is not the only difference between the historical and picontrol simulations? They differ also in terms of changing greenhouse gases? How is this difference going to feedback onto the impacts of deforestation in historical-picontrol? The authors say in the same paragraph that the method assumes that the greenhouse gases affect Tsurf and T2m in neighboring grid points in the same way but that will still cause a constant anomaly in the temperature values owing to the greenhouse gas increase. How is that taken care of in the algorithm so that it is similar to the simulations with MPI-ESM? No further analysis is needed. Only a more clear explanation of the experimental design with the
CMIP5 models will suffice.

4. What type of spatial interpolation technique is used? is it linear or non-linear? Given that the variable field under study could be so heterogeneous (especially Tsurf), it seems that the interpolation technique can have significant impacts on the derived non-local and local fields which can impact the final interpretation of results.

5. What would be the impact of topography and background climate on the interpolated local and non-local signals? Do the authors assume that because an extensive deforestation scenario is considered, the impact of elevation, terrain and background climate on the local and non-local effects is already represented in the deforested simulation?

6. As I understand the deforestation in the MPI-ESM simulations has a regular pattern (3 of 4 grid boxes). Although there is nothing intrinsically wrong in choosing such a deforestation pattern, but there is evidence from previous studies that regular deforestation patterns can trigger climatologically important mesoscale effects. Could the chosen deforestation pattern and any subsequent mesoscale effects have an impact on the simulated local dTsurf? Only an insight from the authors is requested without any additional analysis.

7. Were there any apparent differences in conclusions due to the use of a coupled dynamic ocean model versus the previous studies which used prescribed SSTs? In other words, does a dynamic ocean have a substantial role in deciding the local dTsurf and dT2m responses studied here? I guess a dynamic ocean would be more important for deciding the non-local response. Does this study in conjunction with previous studies throw some light on the role of the ocean in deciding the local and non-local response?

8. Page 6, line 25 - why is the non-local effect cooler and drier?

9. When comparing MPI-ESM results with CMIP5 models the authors point out that the similarities in the results could be due to the similarities in the way models estimate T2m (Page 11, line 14). Could there be other ways to test whether the results obtained
are independent of the model parametrizations? Could this methodology be repeated with some observed/reanalyzed climate time scale global datasets of Tsurf and T2m? Such an analysis need not necessarily be included in the present manuscript but it will be helpful to know author's insights about using observed data with the same methodology. What would be the challenges in such an analysis?

TECHNICAL COMMENTS 1. Page 11 line 7 - remove an extra 'the'

2. Stippling showing significant differences on the difference maps (Figs 1 and 2) would help.

3. Latitude markers on all maps will be helpful.

---

## Referee Comment (RC2) · Anonymous Referee #2 · 13 Nov 2018

In this paper, the authors evaluate how the land cover change response differs for surface temperature relative to air temperature in observations and models. This paper provides important clarification in terms of how these two different temperatures respond to land cover change with observations indicating that changes in surface temperature are roughly twice as strong as they are for air temperature. Models show a varying amount of agreement with the observations. The assessment of local versus nonlocal responses is less developed and is less likely to be the 'final word' on this topic, but I believe the nonlocal results are still worthy of publication. The paper is generally well-written (though it could use another full edit) and the figures are clear. I recommend the paper for publication pending minor revisions as outlined below:

[Figure]

1. Definition of local: The authors use the term local to refer to responses to deforestation within the same grid cell where the deforestation occurs and nonlocal changes to situations where deforestation has effects that extend beyond the deforested region. This is a reasonable definition, but it's worth noting that several recent studies have looked at even more local responses by examining the sub-grid responses within a grid cell (e.g., forested vs cropland). Perhaps it would be helpful to note the difference in definition and to cite a few of these studies (e.g., Malyshev et al., 2015, Schulz et al., 2016, Meier et al., 2018).

2. P.1, Line 17: Not sure I agree with the statement "Much less considered are the climate changes". Researchers have been investigating the climate impacts of deforestation for decades.

3. P.2, Line 34: For clarification, consider changing: "the local effects have to beisolated from the climate model results" to "the local effects need to be disaggregated from the nonlocal effects when analyzing climate model results."

4. P.3, Line 26. Can you provide a rationale for the method of removing forest in 3 out of 4 grid cells? Why not 2 out of 4 or 1 out of 4 or 5 out of 6? Would be helpful to be able to refer to the deforestation map, either as a figure in the main text or as supplemental material.

5. P.5, Line 24. It's not totally clear to me how T2m is derived from S2m using eqn. 1. Equation 1 describes how to calculate Szaero, not S2m.

6. P.5, Line 22: Change "Different functions gamma are used" to "Different functions for gamma are used"

7. P.6, Line 27: An extra "and" in this line.

8. P.6, Line 26: This sentence confused me at first because I had forgotten about the way the global-deforestation run was done (i.e., with 3 out of 4 gridcells deforested). Please clarify. Seems possible that what you mean is actually "large-scale" deforestation rather than global-scale.

9. Figure 1. Seems like some level of significance is needed here or at the very least selection of a color scale that doesn't imply a near-global signal from deforestation (i.e., colors everywhere).

10. P.11, line 5: Should probably note that CCSM4, CESM1-CAM5 and NorESM1 all share the same land model, CLM4).

11. P. 11, line 17. I don't think this is an assumption, this is a result of your analysis. There is no 'assumption' that Tatm does not respond to deforestation.

12. P.12, line 26: Personally, I don't think the comment about carbon cycle feedbacks and the fact that they are non-local is necessary. First, this is stating the obvious. Second, this paper is about biogeophysical impacts.

References

Malyshev, Sergey, et al. "Contrasting local versus regional effects of land-use-change-induced heterogeneity on historical climate: Analysis with the GFDL Earth System Model." Journal of Climate 28.13 (2015): 5448-5469.

Meier, R., Davin, E. L., Lejeune, Q., Hauser, M., Li, Y., Martens, B., et al. (2018). Evaluating and improving the Community Land Model's sensitivity to land cover. Biogeosciences, 15(15), 4731-4757. https://doi.org/10.5194/bg-15-4731-2018

Schultz, N.M., X. Lee, P.J. Lawrence, D.M. Lawrence, L. Zhao, 2016: Assessing the use of sub-grid land model output to study impacts of land cover change. JGR., 121, 6133-6147, DOI: 10.1002/2016JD025094.

---

## Referee Comment (RC3) · Anonymous Referee #3 · 22 Nov 2018

General Comments

The research described in this article addresses an interesting topic – how deforestation affects various measures of temperature, as calculated by global climate models. Overall, I thought the results were well presented, but I had some issues with the way the paper was written, which led to some confusion on my part that required repeated re-reading. Some important ideas were glossed over (e.g., deforestation leading to reduced longwave forcing from above), and I had to infer (possibly incorrectly) some cause-and-effect mechanisms. This will require more interpretation of the results than is presented here.

[Figure]

Specific Comments

Line 13: The 2 effects often associated with deforestation are albedo increases (which cool the surface) and a reduction in transpiration (which reduces the latent heat flux, forcing the sensible heat flux to rise and increasing the surface temperature). Is it the balance between these competing effects that depends on latitude, leading to cooling at some latitudes and warming at others?

Line 24: I'm confused by the way the 'forest world' was created. I understand how forest was placed in areas where it existed in pre-industrial times (but currently does not). Figure 1, however, shows strong local effects in the Sahara and Gobi deserts. Was there any difference in the local forcing at these locations? A map of what the vegetation in the forest world looks like (along with the $\frac{3}{4}$ world) would be helpful.

Line 26: I'm not sure what 'three of four grid boxes' means. Were 3 out of every 4 forested ares randomly selected to be deforested, or was some kind of pattern used?

Line 4: They write 'nonlocal effects strongly depend on the areal extent and spatial distribution of deforestation'. I'm assuming that the deforestation patterns differ among the different climate models, which is why it is impractical to compare nonlocal effects between the different GCMs, correct?

Line 25/26: I'm interpreting this as follows: deforestation leads to a global reduction in temperature and humidity (due to the increases in albedo and decreases in evapotranspiration?), and this leads to more longwave escaping to space and less coming from above. Is this correct, or do changes in cloud cover play a role? If the former, it should be stated more clearly.

Line 27: The pattern of nonlocal effects in Fig. 1 needs some explanation. Why are the eastern Pacific Ocean currents warmer? And why are the forested areas in the Amazon and equatorial Africa warmer? How are they affected by their neighboring, deforested areas? Is this also due to changes in longwave forcing?

Line 4/5: Again, the idea here is that a local effect is propagated remotely by reducing humidity and allowing more IR to escape, correct? And if that is true, what is causing the nonlocal increases of the 3 temperature metrics in the Amazon and equatorial Africa? I'm assuming it is related to the dense forest in these areas, perhaps making the change due to deforestation more pronounced in these locations, but some explanation is needed.

Line 5: Now, it is stated that changes in atmospheric temperature and moisture are affecting longwave radiation. Is deforestation decreasing the global humidity, making the atmosphere more transparent to longwave?

Line 30-35: If 'atmospheric conditions are unstable', why do we not see convective overturning of the atmosphere? This would eliminate the vertical gradient seen in Fig. S3b. Also, how does reducing the roughness length increase instability?

I'm not quite following this explanation for the differences between Figs. S3a and S3b. First, Tsurf is shown to increase during the day and decrease at night. These are linked to changes in stability, and this leads to differences in the way T2m is calculated between night and day with Monin-Obukhov theory and Eq. 2. What is missing is an explanation of the changes in Tsurf, why they differ between day and night, and why the changes vary with latitude (Fig. 2). Are they related to changes in albedo, in evapotranspiration, or both? This seems to be the key driver for the local changes, and ultimately the nonlocal changes as well.

Additionally, invoking the parameterization in Eq. 2 as the explanation of why the T2m

values don't change as much as the Tsurf values is not really explaining why it is happening. Exactly what physical mechanism is causing the 2m temperature to vary less?

Finally, this explanation of differing responses between Tsurf and T2m in summer and during the day is ultimately the reason that these 2 variables look different in the annual averages in Fig. 1, correct? And the way that local changes in Tsurf vary with latitude in Fig. 1 are because the changes in Tmin at the surface dominate at high norther latitudes, while the changes in Tmax dominate elsewhere, correct?

Technical Corrections

Line 5: The acronym MPI-ESM should be spelled out here.

Line 7: The phrase 'effects affect' is awkward, and should be revised.

Line 11: It was already established that the authors were using the MPI-ESM, so what is this 'inter-model comparison' they mention now? A sentence explaining that existing model data from multiple GCMs was examined for comparison to the MPI-ESM results is needed.

Line 15: The 'wide range of climate models' needs more context. As in the abstract, a sentence explaining the idea should suffice.

Line 27: The first sentence of Section 2.3 is confusing and should be rewritten. The phrase 'In order to. . ..other climate models,' is not needed, since it just states the same idea in the rest of the sentence.

Line 6: Change 'deforestation in the difference' to 'deforestation as the difference'.

Figure 1: These are annual means, correct? If so, it should be in the caption.

Line 23: The sentence 'Similarly as in the case. . .' should reference Fig. 2.

Page 11/12

Some information in the Discussions/Conclusions section was already included in the introduction.

<hr />

---

## Referee Comment (RC4) · Anonymous Referee #4 · 22 Nov 2018

This paper "Different response of surface temperature and air temperature to deforestation in climate models" by Johannes Winckler investigated the discrepancy in the temperature response to deforestation between climate model and observations, and how the deforestation impact differs among temperature variables. The question studied here is important to understand the impact of deforestation on temperature. The paper also presents some interesting new findings on this topic. Therefore, I think the paper is suitable for publication in Earth System Dynamics.

Major comments:

I feel the manuscript needs to be edited to improve the language, especially for the use

of preposition like "the", and some sentences are difficult to understand.

According to results of this study, is it possible to establish a relationship to link the impact on surface temperature and on near-surface air temperature to reconcile their differences (a statistical model or the ratio 0.5 found in the paper)?

When analyzing the discrepancy, model uncertainty should be always kept in mind. How the results of this study would be affected by such uncertainty?

Specific comments:

P2 L30: It would be better to also provide the submitted manuscript (Winckler et al., 2018) to reviewers to facilitate the review.

P2 L31: I think these studies compared nearby locations between forest and non-forest or between locations with and without deforestation.

P2 L35: Please specify the different mechanisms here.

P3 L6-8 This sentence needs to be revised for clarity.

P3 L16: add "... from CMIP5."

P4: "The local effects are thus the temperature changes that exceed the nonlocal temperature changes that are obtained by interpolation from nearby non-deforested grid boxes". I don't understand this sentence.

P4 L2.3 Better specify "CMIP5" models

P5 L9: How about the 2m temperature in other models, is it defined in a similar way and thus have the similar problem? As for 2m temperature from observation, is it the 2m above ground (within canopy), or 2m above canopy?

P5 L31: Why only 30 years for the non-local effect? I realized that this is explained later. Maybe some rearrangements can be done for this.

Figure 1: Since the transition latitude from warming to cooling is discussed in the paper,
it would be useful to have a latitudinal averaged temperature response for different temperature variables (or in a separate figure).

P8 L30-35. If the discrepancy is explained this way by Richardson number, it sounds like such discrepancy is a model-dependent artifact instead of actual phenomenon. The discrepancy can be seen in observations (e.g., Baldocchi 2013), suggesting it is not just the Richardson number reason. I guess that the differences in the magnitude of Tmin/Tmax and seasonal responses could play a role because they cancel out each other at the annual mean scale.

P10: L11-13: I don't understand this sentence.

P10 L15: "all but one model show a surface warming locally" this sentence may cause confusion.

P11 L13-14. The 0.5 ratio is an interesting number. Is it applicable to section 3.1?

P12 L23: With the scale of deforestation in reality much smaller than the model simulation, the non-local effect is negligible and the local effect is dominant, this makes the climate model and observation more comparable.

P13 L9-10: There is a possibility that this is due to in climate model uncertainty, we don't know if the model is able to perfectly simulate Tmax response. Model uncertainty needs to be taken into account when making this statement.

A recent paper by Melo-aguilar (2018) might be helpful.

Reference

Baldocchi D, Ma S. How will land use affect air temperature in the surface boundary layer? Lessons learned from a comparative study on the energy balance of an oak savanna and annual grassland. Tellus B. 2013 Melo-aguilar C, González-rouco JF, García-bustamante E, Navarro-montesinos J, Steinert N. Influence of radiative forcing factors on ground – air temperature coupling during the last millenniumŕ: implica-

**ESDD**

**ESDD**

---

## Author Comment (AC1) · 14 Jan 2019

The comment was uploaded in the form of a supplement:
https://www.earth-syst-dynam-discuss.net/esd-2018-66/esd-2018-66-AC1-supplement.pdf

---

## Author Comment (AC2) · 14 Jan 2019

The comment was uploaded in the form of a supplement:
https://www.earth-syst-dynam-discuss.net/esd-2018-66/esd-2018-66-AC2-supplement.pdf

---

## Author Comment (AC3) · 14 Jan 2019

The comment was uploaded in the form of a supplement:
https://www.earth-syst-dynam-discuss.net/esd-2018-66/esd-2018-66-AC3-supplement.pdf

---

## Author Comment (AC4) · 14 Jan 2019

The comment was uploaded in the form of a supplement:
https://www.earth-syst-dynam-discuss.net/esd-2018-66/esd-2018-66-AC4-supplement.pdf

---

## Author Response (AR1)

**Answers to the reviewer comments on the manuscript "Different response of surface temperature and air temperature to the biogeophysical effects of deforestation in climate models" in Earth System Dynamics Discussions**

The following pages contain point-by-point responses to the reviewer comments, separately for each of the four reviewers. The last pages contain a latexdiff pdf of the manuscript, indicating where text was removed (red) or added (blue). We found most of the reviewer comments constructive and thus used them to improve the manuscript accordingly. We hope that our adjustments improved the text.

List of most relevant changes in the manuscript:

- Following the reviewer suggestions, we reorganized the text to improve the flow of the Introduction/Discussions and conclusions sections.
- In the maps in the main text, stippling now indicates statistical significance.
- The English was improved in order to facilitate readability.
- Some information was added in the supplementary material, for instance a map illustrating where deforestation was applied in the MPI-ESM model.
- In the results section we extended the discussion on the mechanisms that may contribute to the simulated change in sign for surface and air temperature in the MPI-ESM model.

**Response to reviewer 1 of the paper entitled**

"Different response of surface temperature and air temperature to deforestation in climate models" Ref.: esd-2018-66

We would thank the reviewer for the time he/she devoted on reviewing the manuscript, and for his/her helpful comments.

Below are the reviewers comments (*bold italic font*) and our responses to each point (normal font). All line numbers that we provide in our responses refer to the revised version of the manuscript in which track changes are not shown.

The original manuscript contained one paper (Winckler et al., 2018) that had not been accepted yet. This manuscript has now been accepted (doi: 10.1029/2018gl080211) and can be made available to the reviewers.

This study endeavors to tease out the differences in the local response to deforestation on surface temperature and near surface air temperature on global scales as derived from an Earth system model and several climate models from the CMIP5 archive. The study uses a clever approach to first estimate nonlocal effects by considering only non-deforested grid points and producing a map of non-local effects by interpolation on deforested grid points. The local effect is then the difference between the total signal (total change in temperature due to deforestation) and the non-local effect. The main findings are that 1) deforestation mainly results in a non-local cooling and drying of the lowest atmospheric level, T2m and Tsurf with warming in the tropical land regions, 2) local effects are more strong and heterogeneous at the surface, 3) in the mid-latitudes the local response to deforestation of Tsurf and T2m can be of different magnitudes and sometimes even opposite. Authors then also try to explain this opposite local response of Tsurf and T2m in the mid-latitudes but the reasoning does not come across very clearly and in my opinion should be revised with details. Overall, the study proposes a potential new statistical method (based on the author's previous work) to address some previously observed differences between the response to defor- estation of Tsurf and T2m. This is a very important research question pertaining to our understanding of the impacts of deforestation on regional climate. This study points out a very important distinction that should be made while interpreting results from datasets of surface temperature versus near surface air temperature. In this regard the study contributes to current knowledge significantly and so is worthy of consideration. However, several important questions regarding the methodology and physical inter-pretation of the results remain which need to be addressed. I would like the authors to comment on my questions with some further analysis if possible/needed as seen fit by the authors. My comments are rather minor but I recommend publication of the study after another round of revisions which I'll be happy to review.

We are happy that the reviewer considers our study to contribute to current knowledge significantly and is worthy of consideration.

1. This is probably outside the scope of the present study but one still questions - what is the mechanism that results in opposite responses of Tsurf and T2m in the mid-latitudes? Can any mechanism be generalized to all such land regions which show opposite responses of dTsurf and dT2m? Probably not because otherwise all land regions between 35 and 55 north as well as south would show the opposite response. The authors do provide an explanation using the model physics and parametrizations (Page 8, line 29) but it is hard to interpret the underlying physics from this argument. Also it is not clear from this argument why such an opposite response will be observed only in the mid-latitudes. I think it will be worthwhile for the authors to include any hypotheses about candidate mechanisms in the manuscript? A bit more explanation in the present manuscript

**is needed if the authors intend to explain this opposite response using the Richardson number, because the argument in its present form is not very clear.**

This paragraph had obviously not been clear yet, see comments of the other reviewers. We now hypothesize in the middle of section 3.1 that 'Part of the difference in the response at the surface and near-surface air could be explained by averaging daytime and nighttime response and averaging the response in different seasons', and in the last sentence of section 3.1 that 'for both variables, the annual mean response then depends on the balance between the daytime and nighttime response, and the balance between the responses in different seasons'. Furthermore, we hypothesize that the way T2m is calculated in the MPI-ESM could cause the opposite response of Tsurf and T2m. In the last two paragraphs of section 3.1, we now discuss separately what one could expect in reality (e.g. stronger near-surface vertical mixing during daytime), and how this is taken into account in the model's calculation, separately for Tmax and Tmin.

2. a) The cross product between dTlocal and dTnonlocal have been neglected based on some analysis by previous studies. But there are other non-local factors that can impact and couple with dTlocal, for example precipitation changes due to circulation changes corresponding to a particular pattern of deforestation can bring about changes in Tsurf via the surface energy budget. These changes will be counted as non-local because they are not a direct consequence of local deforestation. So this component of dTsurf should be accounted for in the non-local dTsurf which is estimated using neighboring grind points. But the neighboring grid points could have an entirely different land cover which could result in a nonrepresentative non-local dTsurf at deforested grid points - because the surface energy balance in these grid points will be different due to different vegetation types.

b) So the effect of such a dTsurf can not be obtained from interpolation from neighboring points. How are such non-local effects from changes in variables other than Tsurf, T2m and Tair considered in the methodology?

c) Do the authors think such cross terms will also be negligible as is the case with dTlocal and dTnonlocal? If so can that be explicitly shown?

a) For our method it does not matter whether other large-scale quantities (besides temperature) undergo changes. For the case of precipitation, we have demonstrated this explicitly in (Winckler et al., 2017a) where in Fig. 3c it can be seen that large-scale deforestation (7 of 8 grid boxes) substantially influences nonlocal changes in precipitation, but this has little influence on the resulting local effects on surface temperature (Fig. 2a and b in Winckler et al. (2017a)).

b) Concerning the problem related to surface heterogeneity: To extract the local signal using our method, also the local conditions (albedo, roughness,...) must vary to a good approximation linearly across the neighboring grid cells. The error that is induced by the horizontal interpolation was assessed in a previous study by two simulations in which the deforestation pattern is shifted, and it was found that in most regions the interpolation error is much smaller than the local effects on surface temperature. (e.g., Fig. D2*e* in Winckler et al. (2017a)).

c) Indeed there are interactions between the local and nonlocal signals. These interactions would be, in our case, lumped with the local effects. We now state this in section 2.1 and explain that the interactions were found to be small across a wide range of deforestation scenarios (Winckler et al., 2017a).

3. Page 6, line 7- I hope I understand this correctly – so land cover change is not the only difference between the historical and picontrol simulations? They differ also in terms of changing greenhouse gases? How is this difference going to feedback onto the impacts of deforestation in historicalpicontrol? The authors say in the same paragraph that the method assumes that the greenhouse gases affect Tsurf and T2m in neighboring grid points in the same way but that will still cause a constant anomaly in the temperature values owing to the greenhouse gas increase. How is that taken care of in the algorithm so that it is similar to the simulations with MPI-ESM? No further analysis is needed. Only a more clear explanation of the experimental design with the CMIP5 models will suffice.

We are sorry that this was not clear in the original version. In section 2.3, we added the following: 'Linear regressions are fitted between temporal changes in temperature (the so-called predictand) and forest cover change (the so-called predictor) within a spatially moving window encompassing  $5 \times 5$  model grid boxes. In the center of this moving window, the local effects are then defined as the temperature change for a hypothetical conversion of 100% forest into 100% open land (given by the slope of the regression) and is by construction largely independent of the changes due to the nonlocal greenhouse gas forcing and nonlocal deforestation effects (given by the *y*-intercept of the regression).'

The greenhouse gas forcing is a nonlocal forcing and therefore removed by this method. We think it is appropriate to implicitly put the CO2 forcing into the nonlocal effects because in Fig. 3 (for which this method is applied) we focus on the local effects only.

4. What type of spatial interpolation technique is used? is it linear or non-linear? Given that the variable field under study could be so heterogeneous (especially Tsurf), it seems that the interpolation technique can have significant impacts on the derived non-local and local fields which can impact the final interpretation of results.

We now state in section 2.1 that we use bi-linear interpolation. We are aware that the interpolation of both local and nonlocal effects may cause an interpolation error, but this interpolation error is comparably small in most areas (see Winckler et al. (2017a), Figs. D2 and D3 e) and f) for the interpolation errors in comparable deforestation scenarios but only 30-year simulations). Furthermore, we think that an interpolation error would affect both dTsurf and dT2m in a similar way and thus would not affect our conclusions regarding the *difference* between dTsurf and dT2m.

5. What would be the impact of topography and background climate on the interpolated local and non-local signals? Do the authors assume that because an extensive deforestation scenario is considered, the impact of elevation, terrain and background climate on the local and non-local effects is already represented in the deforested simulation?

Due to the heterogeneous topography of the land surface, the horizontal interpolation may introduce some error which is generally small (see answer to comment 4). A changing background climate can influence the local effects of deforestation to some extent (see Winckler et al. (2017b) for the change from present-day to RCP8.5 background climate). The only change in background climate in our simulations is caused by nonlocal effects; this change is substantially weaker than the greenhouse-gas-induced change in RCP8.5, so we think that local deforestation effects are not substantially influenced by changing background climate in our simulations.

6. As I understand the deforestation in the MPI-ESM simulations has a regular pattern (3 of 4 grid boxes). Although there is nothing intrinsically wrong in choosing such a deforestation pattern, but there is evidence from previous studies that regular deforestation patterns can trigger climatologically important mesoscale effects. Could the chosen deforestation pattern and any subsequent mesoscale effects have an impact on the simulated local dTsurf? Only an insight from the authors

**is requested without any additional analysis.**

We think that such mesoscale effects are substantially smaller than local/nonlocal dTsurf itself. First, if changes in mesoscale circulations were important (e.g., increased convection over forest, increased subsidence over grass), we would for example expect a reduction in precipitation for the local effects and an increase in precipitation in the nonlocal effects, or the other way round. But this does not seem to be the case; Precipitation is reduced both locally and non-locally (Winckler et al., 2017a). Second, if changes in mesoscale circulations were important for local dTsurf, we would also expect to see the regular spatial pattern both in the local and nonlocal effects. This does not seem to be the case: In a deforestation scenario with only 1 of 8 grid boxes are deforested (Fig. D1b in Winckler et al. (2017a)), the nonlocal effects are equally strong in grid boxes that are directly adjacent to a deforestation grid box and grid boxes that are only surrounded by other no-deforestation grid boxes.

7. Were there any apparent differences in conclusions due to the use of a coupled dynamic ocean model versus the previous studies which used prescribed SSTs? In other words, does a dynamic ocean have a substantial role in deciding the local dTsurf and dT2m responses studied here? I guess a dynamic ocean would be more important for deciding the non-local response. Does this study in conjunction with previous studies throw some light on the role of the ocean in deciding the local and non-local response?

The reviewer is right that a dynamic ocean influences mainly the nonlocal effects: in accordance with (Davin and de Noblet-Ducoudre, 2010), most ocean regions show a slight deforestation-induced cooling following large-scale deforestation (their Fig. *3a*, our Fig. 1) which probably feeds back on the background climate over land regions and causes the nonlocal effects over land to be slightly more cooling/less warming compared to the nonlocal effects in simulations with prescribed sea surface temperatures (see (Winckler et al., 2017a) Fig. D2 and D3 b) for comparable deforestation scenarios). This change in background climate could potentially also affect the local effects. However, also with an interactive ocean the local effects are largely insensitive to the areal extent of deforestation (Winckler et al., 2018), suggesting that the change in background climate, caused even by large-scale deforestation, is too small to substantially affect the local effects on surface temperature.

**8. Page 6, line 25 – why is the non-local effect cooler and drier?**

The first paragraph of section 3.1 now contains a more detailed explanation. The atmosphere becomes cooler and drier because locally deforestation reduces the input of sensible and latent heat from the surface into the atmosphere, see also (Winckler et al., 2018).

9. When comparing MPI-ESM results with CMIP5 models the authors point out that the similarities in the results could be due to the similarities in the way models estimate T2m (Page 11, line 14). Could there be other ways to test whether the results obtained are independent of the model parametrizations? Could this methodology be repeated with some observed/reanalyzed climate time scale global datasets of Tsurf and T2m? Such an analysis need not necessarily be included in the present manuscript but it will be helpful to know author's insights about using observed data with the same methodology. What would be the challenges in such an analysis?

It would be desirable to repeat the analysis with observation-based gridded datasets. However, we are not aware of any such dataset for T2m, which in reanalysis datasets is based on a model or semi-empirical formulas. Thus, the results would again depend on how T2m is calculated for day/night conditions and over different vegetation types in these formulas.

**10. TECHNICAL COMMENTS**

Page 11 line 7 – remove an extra 'the'

Done.

**11. Stippling showing significant differences on the difference maps (Figs 1 and 2) would help.**

We added stippling for grid boxes where results are not significant at a 5% level according to a student t-test accounting for lag-1 autocorrelation.

**12. Latitude markers on all maps will be helpful.**

We added latitude markers on all maps.

**References**

- Davin, E. L. and de Noblet-Ducoudre, N. (2010). Climatic impact of global-scale deforestation: radiative versus nonradiative processes. *Journal of Climate*, 23(1):97–112.
- Winckler, J., Reick, C. H., Lejeune, Q., and Pongratz, J. (2018). Nonlocal effects dominate the global mean surface temperature response to the biogeophysical effects of deforestation. *Geophysical Research Letters*.
- Winckler, J., Reick, C. H., and Pongratz, J. (2017a). Robust identification of local biogeophysical effects of land-cover change in a global climate model. *Journal of Climate*, 30(3):1159–1176.
- Winckler, J., Reick, C. H., and Pongratz, J. (2017b). Why does the locally induced temperature response to land cover change differ across scenarios? *Geophysical Research Letters*, 44:3833–3840.

**Response to reviewer 2 of the paper entitled**

"Different response of surface temperature and air temperature to deforestation in climate models" Ref.: esd-2018-66

We would thank the reviewer for the time he/she devoted on reviewing the manuscript, and for his/her helpful comments.

Below are the reviewers comments (*bold italic font*) and our responses to each point (normal font). All line numbers that we provide in our responses refer to the revised version of the manuscript in which track changes are not shown.

The original manuscript contained one paper (Winckler et al., 2018) that had not been accepted yet. This manuscript has now been accepted (doi: 10.1029/2018gl080211) and can be made available to the reviewers.

In this paper, the authors evaluate how the land cover change response differs for surface temperature relative to air temperature in observations and models. This paper provides important clarification in terms of how these two different temperatures respond to land cover change with observations indicating that changes in surface temperature are roughly twice as strong as they are for air temperature. Models show a varying amount of agreement with the observations. The assessment of local versus nonlocal responses is less developed and is less likely to be the 'final word' on this topic, but I believe the nonlocal results are still worthy of publication. The paper is generally well-written (though it could use another full edit) and the figures are clear. I recommend the paper for publication pending minor revisions as outlined below:

We are happy that the reviewer thinks our results are worthy of publication.

1. Definition of local: The authors use the term local to refer to responses to deforestation within the same grid cell where the deforestation occurs and nonlocal changes to situations where deforestation has effects that extend beyond the deforested region. This is a reasonable definition, but it's worth noting that several recent studies have looked at even more local responses by examining the sub-grid responses within a grid cell (e.g., forested vs cropland). Perhaps it would be helpful to note the difference in definition and to cite a few of these studies (e.g., Malyshev et al., 2015, Schulz et al., 2016, Meier et al., 2018).

It is indeed important to clarify differences to these previous studies. In the last paragraph of section 2.1, we now cite these studies and shortly discuss differences in the definitions of local effects.

2. P.1, Line 17: Not sure I agree with the statement 'Much less considered are the climate changes'. Researchers have been investigating the climate impacts of deforestation for decades.

We corrected the respective text. Now: 'In addition, changes in forest cover can cause a warming or cooling,...'

3. P.2, Line 34: For clarification, consider changing: 'the local effects have to be isolated from the climate model results' to 'the local effects need to be disaggregated from the nonlocal effects when analyzing climate model results.'

We changed the text accordingly (with 'separated' instead of 'disaggregated').

4. P.3, Line 26. Can you provide a rationale for the method of removing forest in 3 out of 4 grid cells? Why not 2 out of 4 or 1 out of 4 or 5 out of 6? Would be helpful to be able to refer to the deforestation map, either as a figure in the main text or as supplemental material.

We now acknowledge (last paragraph of section 2.1) that the choice of 3 of 4 grid cells is to some extent arbitrary, but the local effects within a grid cell are largely insensitive to this choice (Winckler et al., 2017a). We now provide the deforestation map in Fig. S1.

5. P.5, Line 24. It's not totally clear to me how T2m is derived from S2m using eqn. 1. Equation 1 describes how to calculate Szaero, not S2m.

We now state that  $z_{aero} = 2m + d + z_0$  has to be used in equation 1 in order to derive T2m.

6. P.5, Line 22: Change 'Different functions gamma are used' to 'Different functions for gamma are used'. Line 27: An extra 'and' in this line.

Thanks, corrected.

7. P.6, Line 26: This sentence confused me at first because I had forgotten about the way the globaldeforestation run was done (i.e., with 3 out of 4 gridcells deforested). Please clarify. Seems possible that what you mean is actually 'large-scale' deforestation rather than global-scale.

We changed 'global-scale' to 'large-scale'.

8. Figure 1. Seems like some level of significance is needed here or at the very least selection of a color scale that doesn't imply a near-global signal from deforestation (i.e., colors everywhere).

We added stippling to indicate where results are not significant at a 5% level according to a student t-test accounting for lag-1 auto-correlation (Zwiers and von Storch, 1995).

9. P.11, line 5: Should probably note that CCSM4, CESM1-CAM5 and NorESM1 all share the same land model, CLM4).

I added this remark, thanks.

10. P. 11, line 17. I don't think this is an assumption, this is a result of your analysis. There is no 'assumption' that Tatm does not respond to deforestation.

I changed the phrasing, thanks.

11. 12. P.12, line 26: Personally, I don't think the comment about carbon cycle feedbacks and the fact that they are non-local is necessary. First, this is stating the obvious. Second, this paper is about biogeophysical impacts.

This statement may be obvious for experts on deforestation effects in climate models, but possibly not for the broad audience of ESD. We would like to keep this statement to make non-experts aware that the biogeophysical effects (for which dT seems to differ especially for the local effects) are only one part of the deforestation effects, and that the other part, the carbon effects, can be expected to act essentially nonlocally.

Malyshev, Sergey, et al. "Contrasting local versus regional effects of land-use-change- induced heterogeneity on historical climate: Analysis with the GFDL Earth System Model." Journal of Climate 28.13 (2015): 5448-5469.

Meier, R., Davin, E. L., Lejeune, Q., Hauser, M., Li, Y., Martens, B., et al. (2018). Evaluating and improving the Community Land Model's sensitivity to land cover. Bio-geosciences, 15(15), 4731-4757. https://doi.org/10.5194/bg-15-4731-2018

Schultz, N.M., X. Lee, P.J. Lawrence, D.M. Lawrence, L. Zhao, 2016: Assessing the use of subgrid land model output to study impacts of land cover change. JGR., 121, 6133-6147, DOI: 10.1002/2016JD025094.

b) Page 2 Line 26: I'm not sure what 'three of four grid boxes' means. Were 3 out of every 4 forested ares randomly selected to be deforested, or was some kind of pattern used?

a) We now provide a map (Fig. S1) showing which grid boxes were deforested (in a regular spatial pattern), The map also shows the fraction of vegetated areas in these grid box. In the forest world, forest is prescribed on all vegetated areas (e.g., 100% in most grid boxes where forest is present today,

but also on present-day grasslands) but close to 0% in areas with sparse present-day vegetation cover such as the Sahara or Gobi deserts. In the deforestation simulation, forests in the vegetated areas is replaced by 100% grasslands. The reviewer is right that there is some local change in surface temperature in the deserts although only a small fraction of the grid box was deforested. We hypothesize that this comparably large signal could be caused by the non-linearity in the response of surface temperature to changes in forest cover within a grid box (Winckler et al., 2017) (meaning that for low initial forest cover, small changes can have a large effect).

b) For the deforestation, a regular spatial pattern was used. In section 2.1 we now refer to the newly added deforestation map, see Fig. S1

4. Page 6 Line 4: They write 'nonlocal effects strongly depend on the areal extent and spatial distribution of deforestation'. I'm assuming that the deforestation patterns differ among the different climate models, which is why it is impractical to compare nonlocal effects between the different GCMs, correct?

We removed this point because it is not essential and it would require substantially more explanation to clarify.

5. Page 6 Line 25/26: I'm interpreting this as follows: deforestation leads to a global reduction in temperature and humidity (due to the increases in albedo and decreases in evapotranspiration?), and this leads to more longwave escaping to space and less coming from above. Is this correct, or do changes in cloud cover play a role? If the former, it should be stated more clearly.

Page 6 Line 27: The pattern of nonlocal effects in Fig. 1 needs some explanation. Why are the eastern Pacific Ocean currents warmer? And why are the forested areas in the Amazon and equatorial Africa warmer? How are they affected by their neighboring, deforested areas? Is this also due to changes in longwave forcing?

Page 8 Line 4/5: Again, the idea here is that a local effect is propagated remotely by reducing humidity and allowing more IR to escape, correct? And if that is true, what is causing the nonlocal increases of the 3 temperature metrics in the Amazon and equatorial Africa? I'm assuming it is related to the dense forest in these areas, perhaps making the change due to deforestation more pronounced in these locations, but some explanation is needed.

Page 8 Line 5: Now, it is stated that changes in atmospheric temperature and moisture are affecting longwave radiation. Is deforestation decreasing the global humidity, making the atmosphere more transparent to longwave?

These four reviewer comments refer to the mechanisms and spatial patters for the nonlocal effects. The first paragraph in section 3.1 now provides a slightly more detailed explanation. At deforested locations, the input of latent and sensible heat into the atmosphere is reduced. This leads to a drier and cooler atmosphere, and this in turn reduces the longwave incoming radiation, also at locations that were not deforested (nonlocal effects). This explanation, as well as maps of the local changes in latent/sensible heat and nonlocal shortwave/longwave incoming radiation are included in the Supplementary Figures of Winckler et al. (2018) which is now also available to the reviewers. In the Amazon and in equatorial Africa the reduction of longwave incoming radiation is overcompensated by an increase in shortwave incoming radiation due to a reduction in cloud cover (see Figure 1 below).

We feel that a more detailed analysis of the nonlocal effects, i.e. changes in the ocean circulation, goes beyond the scope of the current study.

Figure 1: Deforestation-induced annual mean changes in cloud cover in the '3/4' simulation using the MPI-ESM. The locations where cloud cover decreases for the nonlocal effects are co-located with regions with a nonlocal increase of incoming radiation (Fig. S6 in Winckler et al. (2018)).

6. Page 8 Line 30-35:

a) If 'atmospheric conditions are unstable', why do we not see convective overturning of the atmosphere? This would eliminate the vertical gradient seen in Fig. S3b.

b) Also, how does reducing the roughness length increase instability?

c) I'm not quite following this explanation for the differences between Figs. S3a and S3b. First, Tsurf is shown to increase during the day and decrease at night. These are linked to changes in stability, and this leads to differences in the way T2m is calculated between night and day with Monin-Obukhov theory and Eq. 2. What is missing is an explanation of the changes in Tsurf, why they differ between day and night, and why the changes vary with latitude (Fig. 2). Are they related to changes in albedo, in evapotranspiration, or both? This seems to be the key driver for the local changes, and ultimately the nonlocal changes as well.

d) Additionally, invoking the parameterization in Eq. 2 as the explanation of why the T2m values don't change as much as the Tsurf values is not really explaining why it is happening. Exactly what physical mechanism is causing the 2m temperature to vary less?

e) Finally, this explanation of differing responses between Tsurf and T2m in summer and during the day is ultimately the reason that these 2 variables look different in the annual averages in Fig. 1, correct? And the way that local changes in Tsurf vary with latitude in Fig. 1 are because the changes in Tmin at the surface dominate at high norther latitudes, while the changes in Tmax dominate elsewhere, correct?

a) Indeed, during daytime we would expect potential air temperature to be similar for T2m and Tatm. We suggest two potential reasons for the temperature gradient in Fig. 3b): First, even without a vertical gradient in potential temperature we would expect a gradient in actual temperature due to the lapse rate. Second, the calculation of  $T_{2m}$  is based on semi-empirical formulas and does not explicitly account for the input of sensible heat from the surface or vertical mixing within the atmosphere.

b) In the new version of the paragraph, we don't mention surface roughness because this was obviously confusing. We hypothesize that reducing roughness length could increase daily maximum surface temperature by reducing the ability of the surface to transfer latent and sensible energy into the atmosphere. The surface would have to warm up more (compared to a rougher surface) in order to get rid of energy via longwave outgoing radiation (Stefan-Boltzmann-law). Consequently also the

gradient between the maximum temperature at the surface and the atmosphere could increase. c) Why surface temperature responds differently for daytime and nighttime is an interesting question, but has already been investigated in previous studies (e.g., Schultz et al., 2017). We now refer to the previous studies also in these sentences.

d) In the last two paragraphs of section 3.1 we now provide a physical explanation of what would intuitively be expected and how the differences in near-surface stability between day and night are taken into account in the models' calculation of T2m.

e) We now hypothesize that part of the difference between the response of Tsurf and T2m can be explained by differences during daytime/nighttime and during the different seasons. We now also discuss in section 3.1 that the way T2m is calculated in the model may be important for explaining why the response of T2m and Tsurf can differ even e.g. for Tmax in JJA, see Fig. S9, and that the annual mean response depends on the balance between the daytime and nighttime response, and the balance between the responses in different seasons.

7. Technical Corrections

Page 1 Line 5: The acronym MPI-ESM should be spelled out here. Line 7: The phrase 'effects affect' is awkward, and should be revised.

Corrected, thanks!

8. Line 11: It was already established that the authors were using the MPI-ESM, so what is this 'inter-model comparison' they mention now? A sentence explaining that existing model data from multiple GCMs was examined for comparison to the MPI-ESM results is needed. Page 3 Line 15: The 'wide range of climate models' needs more context. As in the abstract, a sentence explaining the idea should suffice.

We added such a sentence in the abstract and the introduction.

9. Page 5 Line 27: The first sentence of Section 2.3 is confusing and should be rewritten. The phrase 'In order to. . ..other climate models,' is not needed, since it just states the same idea in the rest of the sentence.

Page 6 Line 6: Change 'deforestation in the difference' to 'deforestation as the difference'.

We changed the respective sentences.

10. Page 7 Figure 1: These are annual means, correct? If so, it should be in the caption.

Done.

11. Page 8 Line 23: The sentence 'Similarly as in the case. . .' should reference Fig. 2.

Done.

12. Page 11/12 Some information in the Discussions/Conclusions section was already included in the introduction.

We revised the introduction and discussions/conclusions to avoid repetition of information.

a) I feel the manuscript needs to be edited to improve the language, especially for the use of preposition like 'the', and some sentences are difficult to understand.

b) According to results of this study, is it possible to establish a relationship to link the impact on surface temperature and on near-surface air temperature to reconcile their differences (a statistical model or the ratio 0.5 found in the paper)?

c) When analyzing the discrepancy, model uncertainty should be always kept in mind. How the results of this study would be affected by such uncertainty?

a) The language was improved, we hope that the sentences are now easier to understand.

b) We think that it's a good idea by the reviewer to develop a statistical model to derive the T2m response from the Tsurf response. However, this is a non-trivial task as this ratio may vary by location and season (e.g. Figs. S7 and S8; Fig. 3 is only for DJF/JJA in the northern mid-latitudes) and goes beyond the scope of our study. Although the models seem to agree that the ratio of dT2m:dTsurf is around 1:2 over the studied region and the considered seasons, the exact ratio between the response of T2m and Tsurf is still to some extent model specific (range for JJA: 0.35-0.66 excl. HadGEM2-ES, see Table S1).

c) The reviewer is right that it is important to be aware of model uncertainties. In the last paragraph of section 3.2, we argue that the inter-model differences are large for dTsurf, but smaller for the ratio between dTsurf and dT2m.

**2. P2 L30: It would be better to also provide the submitted manuscript (Winckler et al., 2018) to reviewers to facilitate the review.**

We are sorry about the inconvenience in the first phase of the review process. The manuscript

(Winckler et al., 2018) is now published (doi: 10.1029/2018gl080211) and can be made available to the reviewers.

3. P2 L31: I think these studies compared nearby locations between forest and non-forest or between locations with and without deforestation.

We adjusted the text accordingly.

4. P2 L35: Please specify the different mechanisms here.

We now specify one sentence later that the local effects act predominantly via changes in turbulent heat fluxes, while the nonlocal effects act predominantly via changes in incoming radiation that reaches the surface (Winckler et al., 2018).

5. P3 L6-8 This sentence needs to be revised for clarity.

We removed this sentence which was obviously confusing and did not add much value.

6. P3 L16: add '. . . from CMIP5.'

We adjusted this sentence and included the 'CMIP5'.

7. P4: 'The local effects are thus the temperature changes that exceed the nonlocal temperature changes that are obtained by interpolation from nearby non-deforested grid boxes'. I don't understand this sentence.

We removed this sentence because it was obviously not clear, and it was anyway only meant to be a summary of what was explained above.

8. P4 L2.3 Better specify 'CMIP5' models

The title of section 2.3 is now 'Isolation of local effects across CMIP5 models'.

9. P5 L9: How about the 2m temperature in other models, is it defined in a similar way and thus have the similar problem?

As for 2m temperature from observation, is it the 2m above ground (within canopy), or 2m above canopy?

We think that it is reasonable for climate models to use semi-empirical formulas based on Monin-Obukhov similarity theory (see last paragraph of section 3.2), and thus we expect that also in other models temperature is defined 2m above  $d + z_0$  rather than 2m above the surface or canopy. Concerning the observations, we state in the discussions section (around p. 13, l. 19) that weather stations (i.e. in forest clearings) record temperatures at a height of between 1.2m and 2.0m above

ground level while temperature at Fluxnet sites is typically recorded 2-15m above forest canopies.

10. P5 L31: Why only 30 years for the non-local effect? I realized that this is explained later. Maybe some rearrangements can be done for this.

Section 2.3 is now re-arranged such that first the '30 years' are introduced.

11. Figure 1: Since the transition latitude from warming to cooling is discussed in the paper, it would be useful to have a latitudinal averaged temperature response for different temperature variables (or in a separate figure).

We now provide zonal land averages of the responses of T2m and Tsurf in Fig. S2.

12. P8 L30-35. If the discrepancy is explained this way by Richardson number, it sounds like such discrepancy is a model-dependent artifact instead of actual phenomenon. The discrepancy can be seen in observations (e.g., Baldocchi 2013), suggesting it is not just the Richardson number reason. I guess that the differences in the magnitude of Tmin/Tmax and seasonal responses could play a role because they cancel out each other at the annual mean scale.

The reviewer is right that the magnitude of the Tmin/Tmax and seasonal responses are important to explain the annual mean response of Tsurf and T2m, we added this as the last sentence in section 3.1. However, we think that differences between Tmin/Tmax and between seasonal responses are not the only reason why Tsurf and T2m can respond differently; e.g., even for Tmax in JJA, in some regions T2m and Tsurf can show a different response (Fig. S9.)

We re-wrote the last two paragraphs of section 3.1 (for Tmax and analogously for Tmin) to clarify that there is a plausible mechanism why T2m and Tsurf could respond differently in reality, and how this mechanism is implicitly accounted for in the calculation of T2m in the MPI-ESM.

13. P10: L11-13: I don't understand this sentence. P10 L15: 'all but one model show a surface warming locally' this sentence may cause confusion.

Both sentences were rewritten, we hope they are now more clear.

14. P11 L13-14. The 0.5 ratio is an interesting number. Is it applicable to section 3.1?

As can be seen in Fig. S2, this ratio varies with latitude, even when focusing on annual means. It seems plausible that this ratio may vary also with the considered season.

**15. P12 L23: With the scale of deforestation in reality much smaller than the model simulation, the non-local effect is negligible and the local effect is dominant, this makes the climate model and observation more comparable.**

The reviewer is right that the nonlocal effects in reality are much smaller than in our simulation '3/4'. We now clarify in the caption of Fig. 1 that the shown results refer to this simulation. Furthermore, we added in the text that the nonlocal effects are expected to be large especially in simulations of large-scale deforestation. This does not alter our statement that including the nonlocal effects causes an inconsistency in comparing the models and observations.

16. P13 L9-10: There is a possibility that this is due to in climate model uncertainty, we don't know if the model is able to perfectly simulate Tmax response. Model uncertainty needs to be taken into account when making this statement.

**We replaced this sentence by the following:**

'Our results for the MPI-ESM suggest that the difference between  $T_{2m}$  and  $T_{surf}$  is particularly strong for mean daily maximum temperature (see Fig. 2). Further studies may investigate whether this is also true for other climate models and observation-based data-sets.'

**17. A recent paper by Melo-Aguilar (2018) might be helpful.**

Thanks! The two suggested references are now included in the introduction.

**Different response of surface temperature and air temperature to deforestation in climate models**

Johannes Winckler1,2,3, Christian H. Reick1, Sebastiaan Luyssaert4, Alessandro Cescatti5, Paul C. Stoy6, Quentin Lejeune7,8, Thomas Raddatz1, Andreas Chlond1, Marvin Heidkamp1,2, and Julia Pongratz1,9 1Max Planck Institute for Meteorology, Hamburg, Germany. 2International Max Planck Research School on Earth System Modeling, Hamburg, Germany. 3Current affiliation: Laboratoire des Sciences du Climat et de l'Environnement, LSCE/IPSL, CEA-CNRS-UVSQ, Université Paris-Saclay, Gif-sur-Yvette, France. 4Vrije Universiteit Amsterdam, Faculty of Science, Amsterdam, the Netherlands. 5European Commission, Joint Research Centre, Institute for Environment and Sustainability, Ispra, Italy. 6Department of Land Resources and Environmental Sciences, Montana State University, Bozeman, MT, USA. 7Institute for Atmospheric and Climate Science, ETH-Zürich, Switzerland. 8Current affiliation: Climate Analytics, Berlin, Germany. 9Ludwig-Maximilians-Universität München, Munich, Germany. **Correspondence:** Johannes Winckler(johannes.winckler@lsce.ipsl.fr)

Abstract. Deforestation affects temperatures at the land surface and higher up in the atmosphere Temperatures changes following deforestation are well documented but considerable uncertainty remains concerning their precise value. Part of this uncertainty may be caused by the fact that different methods used to quantify temperature changes, i.e., satellite-based observations, in-situ observations, and climate models, consider different temperatures. Satellite-based observations typically register deforestation-

- 5 induced changes in surface temperature, in-situ observations register changes in near-surface air temperature, and climate models simulate changes in both temperatures and the temperature of the lowest atmospheric layer. Yet a focused analysis of how these variables respond differently to deforestation is missing. Here, this is investigated consistent comparison of the response to deforestation for different temperature variables was missing. In this study the effects of deforestation on surface temperature, near-surface air temperature and lower atmospheric temperature are compared by analyzing the biogeophysical
- 10 temperature effects of large-scale deforestation in the elimate model Max-Planck-Institute Earth System Model (MPI-ESM, ) separately for local effects (which are only apparent at the location of deforestation) and nonlocal effects (which are also apparent elsewhere). While the nonlocal effects affect influence the temperature of the surface and lowest atmospheric layer equally, the local effects mainly affect the temperature of the surface. In agreement with observation-based studies, the local effects on surface and near-surface air temperature respond differently in the MPI-ESM, both concerning the magnitude of lo-
- 15 cal temperature changes and the latitude at which the local deforestation effects turn from a cooling to a warming (at 45-55° N for surface temperature and around 35° N for near-surface air temperature). An Subsequently, our single-model results are compared to model data from multiple climate models. This inter-model comparison shows that in the northern mid latitudes, both for summer and winter, near-surface air temperature is affected by the local effects only about half as much compared to strongly as surface temperature. Thus, studies. This study shows that the choice of temperature variable has a considerable effect

[revised manuscript text omitted]

- 5 framework, and thus elimate models are suitable rendering climate models suitable tools to investigate this question. Both the air and surface temperature can be influenced are affected by local and nonlocal biogeophysical effects of deforestation. We define local effects as effects that are only apparent in deforested locations and nonlocal effects as effects that are also apparent in non-deforested locations (Methods and Winckler et al., 2017)(see Sect. 2.1 and Winckler et al., 2017). Local effects can for example be caused by a redistribution of heat between the surface and the atmosphere (e.g., Vanden Broucke
- 10 et al., 2015) while the nonlocal effects can be caused by advection (Winckler et al., 2018) or by changes in global circulation (Swann et al., 2012; Devaraju et al., 2015; Lague and Swann, 2016)<del>or advection (Winckler et al., 2018). Here, we consider</del> . In this study, local and nonlocal effects are analyzed separately for three reasons. First, the difference between local and nonlocal effects matters for decision makers: the local effects may be relevant for policies that aim at adapting to a warming climate locally because they link the climate effects to the areas where policies are implemented (Duveiller et al., 2018). The
- 15 nonlocal effects are also relevant for international policies that aim at mitigating global climate change because the nonlocal effects may dominate the global mean biogeophysical temperature response to deforestation (Winckler et al., 2018). Second, the observation-based data-sets only record the local effects when comparing temperature between nearby locations with and without forest, or between locations with and without deforestation. The nearby locations share the same background climate, and thus the nonlocal effects cancel out when temperature differences between the locations are considered (Lee et al., 2011; Li
- 20 et al., 2015; Alkama and Cescatti, 2016; Duveiller et al., 2018). For a consistent comparison to observation-based data-sets, the local effects have to be isolated from the need to be separated from the nonlocal effects when analyzing climate model results. The third reason to consider local and nonlocal temperature changes separately is that different mechanisms trigger local and nonlocal temperature changes (Winckler et al., 2017). If surface and air temperature respond differently to deforestation, it is unclear whether this difference arises from the mechanisms that trigger the local temperature changes , the (predominantly via)
- 25 changes in the turbulent heat fluxes (Winckler et al., 2018)), mechanisms that trigger the nonlocal temperature changes , or (predominantly via the incoming radiation that reaches the surface (Winckler et al., 2018)), or from both. A separate analysis of local and nonlocal temperature changes facilitates an investigation of the mechanisms that may cause a different response of surface and air temperature to deforestation.

Here, we investigate how deforestation in the MPI-ESM climate model affects surface and air temperature differently and analyze this separately for the local and nonlocal effects. Thus, we emulate the deforestation effects on surface temperature as

estimated from satellite data and

A previous study contrasted the response of surface temperature only with the response of near-surface air temperature as estimated from in-situ measurements within a consistent framework. In a previous study it was noted that and found mainly differences between surface and air temperature response differ mainly for the local effects (Appendix C in Winckler et al.,

35 2017). We go beyond this previous study by additionally analyzing the effects on temperature in the lowest atmospheric layer

and by using simulations with an interactive ocean because this is essential to capture the full climate which enables us to better capture the nonlocal temperature effects of deforestation (Davin and de Noblet-Ducoudre, 2010). This previous study (Winckler et al., 2017) contrasted the response of surface temperature only with the response of near-surface air temperature, while here we additionally analyze the effects on temperature in the lowest atmospheric layerfrom the surface to the lower

- 5 atmosphere (Davin and de Noblet-Ducoudre, 2010). To further analyze the mechanisms that are responsible for differences in these three temperature variables, we investigate the local effects separately for the response in mean daily minimum and maximum temperature. To test the robustness of our results for this particular climate model, we compare the simulation results of the MPI-ESM to existing

---

## Referee Report (RR1)

I have read the revised version of the article, and noted the changes made in response to my review of the previous version. I'm still finding it confusing to read and fully comprehend, however, and therefore recommend more changes be made before publication. Among other things, I am still not understanding the physical mechanism that the authors propose to explain how a surface increase in $T_{max}$ due to deforestation can produce a cooling at 2m above the surface. Given the focus on differences between $\Delta T_{surf}$ and $\Delta T_{2m}$, this seems to be a major part of the article.

P. 3, Line 20: This may be a style issue, but the acronym MPI-ESM should be spelled out in the introduction, not just in the abstract.

P. 3, line 20: A paragraph should comprise more than one sentence.

P. 4, line 28: Should read '…allows **us** to…'.

P. 5, line 7: 'following' is misspelled.

P. 5, line 5/6: Using '$T_{atm}$' and 'atmospheric temperature' interchangeably is confusing, as is using '$T_{2m}$', 'air temperature', and 'near surface temperature'. This is especially true when, for example, the text describes 'near surface temperature' and refers to a figure in which that same variable is now called '$T_{2m}$'. Please change them to be consistent.

P. 6, Eq. 4: $s_{atm}$ is the dry static energy at $T_{atm}$, right?

P. 9, Line 24: I don't quite understand the sentence 'Part of the difference….'. The responses of $T_{surf}$ and $T_{2m}$ are different with (Fig. 2) or without (Fig. 1) averaging, so I don't see how averaging could explain the different responses.

P. 9, Line 29: Fig. S2 is for annual mean, and doesn't say anything about daytime or 'lowest atmospheric layer' (that's $T_{atm}$, correct?). Should this refer to Fig. 2?

P. 9, line 30: 'deforestation further increases surface temperature (Fig. 1)'. Should this refer to Fig. 2?

P. 9: Fig. 2 shows the maps for DJF and JJA, but you don't explain why they look as they do (as is done for $T_{min}$ and $T_{max}$). I'm assuming that the $T_{max}$

effect ($\Delta T_{surf} > 0$, $\Delta T_{2m} < 0$) demonstrated in Fig. S5b explains the areas where $T_{surf}$ and $T_{2m}$ are different signs in JJA, while the Tmin effect ($\Delta Tsurf < 0$, $\Delta T_{2m} < 0$) explains why $T_{2m}$ and $T_{surf}$ look similar in DJF, correct?

P. 11, line 19: I don't see how Fig. S6 shows what is described in that sentence. MPI-ESM-LR represents a completely different simulation than the one done by the authors with MPI-ESM, and it's hard to make out any details on these plots in any event.

P. 9, Line 31: I am still not satisfied with the explanation of how $T_{2m}$ can cool for $T_{max}$ while $T_{surf}$ rises. The article again invokes the scheme used to interpolate $T_{2m}$ using $T_{surf}$ and $T_{atm}$, but there is still no real explanation as to how in reality an increase in surface heating would not lead to an increase in both vertical mixing and $T_{2m}$, especially over a distance as small as 2m. Fig. 2 implies that the $\Delta T_{surf} > 0$, $\Delta T_{2m} < 0$ effect in the midlatitudes exists for JJA, but the other climate models do not show this (Fig. 3). You refer to the work of Meier et al. as having seen such an effect, but that too was only seen in a model, and they mention that observations contradict this. This is being put forth as a major reason that $\Delta T_{surf}$ and $\Delta T_{2m}$ differ, so I think a better explanation is needed.

Figure S1: I have several questions about this figure:

1. The colored areas represent the potential vegetation from the Pongratz study, correct?

2. Your response to me (#3) seems to imply that forest world was created by simply replacing all current grasslands with forest, and deforested world was created by replacing current forest with grasslands. On page 4, line 8, however, your reference to the Pongratz study implies that a recreation of pre-industrial vegetation was done, which would be something different.

3. What do the dots mean? Are they centers of the grid boxes, or do they have something to do with the ¾ deforestation pattern?

4. In the Figure S1 caption: "The 100% forest is replaced by 100% grasslands…". This is only true for 3 of every 4 grid boxes, correct?

---

## Author Response (AR2)

**Response to the re-review of the paper entitled**
**"Different response of surface temperature and air temperature to deforestation in climate models"**
Ref.: esd-2018-66

We thank the reviewers and the editor for the time they devoted on reviewing the revised version of the manuscript, and for the helpful comments.

Below are the reviewers comments (***bold italic font***) and our responses to each point (normal font). The provided line numbers refer to the revised version of the manuscript in which track changes are not shown.

**Comments from the editor**

*Dear Authors, thank you for submitting your revised manuscript. This was reviewed by 3 referees. The recommendations are generally positive and I think the paper is very close. But it needs a little bit more work. First, one reviewer still has problems understanding the "physical mechanism that the authors propose to explain how a surface increase in Tmax due to deforestation can produce a cooling at 2m above the surface". I agree with the reviewer's concern. Second, the abstract and the conclusions need more clarity. I understand that you want to focus on the differences between $T_{surf}$, $T_{2m}$ and $T_{atm}$ but it will help the readers if you first establish a baseline. For example you may want to start of with something like "deforestation leads to a warming in $T_{surf}$ in the tropics but a cooling in the higher latitudes" and then discuss the differences. I hope you will address these issues in your revised manuscript.*

Concerning the lack of a physical explanation for the counter-intuitive behavior of $T_{2m}$ compared to $T_{surf}$: We appreciate your and reviewers insistence to clarify this issue on physical grounds, and not only in terms of abstract equations. After re-thinking this issue, we now provide a physical explanation – please see our answers to comments 1 and 12 below. With this explanation we think our results now gain sufficient credit against the worry that they are simply an artifact of the model formulation. We consider this as a major improvement thanks to your intervention.
Concerning the abstract and the conclusions: we agree that the abstract is not sufficiently succinct and think that this impression partly comes from the lengthy introductory sentences and missing statements about the direction of temperature changes (also in the conclusions). We tried to remedy this in the revised version.

**Reviewer #3, report #2**

1. *I have read the revised version of the article, and noted the changes made in response to my review of the previous version. I'm still finding it confusing to read and fully comprehend, however, and therefore recommend more changes be made before publication. Among other things, I am still not understanding the physical mechanism that the authors propose to explain how a surface increase in $T_{max}$ due to deforestation can produce a cooling at 2m above the surface. Given the focus on differences between $\Delta T_{surf}$ and $\Delta T_{2m}$ , this seems to be a major part of the article.*

   We now explain this counter-intuitive behavior of $T_{2m}$, by noting that by the increased surface temperature, the well mixed zone in the boundary layer not only extends to larger heights, but also extends further down. Accordingly, the cooler air from above mixes further down, and if this affects heights below 2m, $T_{2m}$ is lowered. Moreover we want to note that the interpolation used to derive $T_{2m}$ is based in Geleyn (1988), who derives the interpolation formula for $T_{2m}$ consistently with the Monin-Obukhov similarity theory. Hence, this is the best we have and fully state of the art and

probably the way how $T_{2m}$ is calculated in most climate models. Further we completely agree that we report here only model results, but with the argument given now, this counter-intuitive result is getting plausibility of not simply being an artifact.

2. **P. 3, Line 20: This may be a style issue, but the acronym MPI-ESM should be spelled out in the introduction, not just in the abstract.**

   We now spell out the acronym MPI-ESM also in the introduction.

3. **P. 3, line 20: A paragraph should comprise more than one sentence.**

   The sentence is now the first sentence of the last paragraph.

4. **P. 4, line 28: Should read '...allows us to...'.**

   Corrected, thanks.

5. **P. 5, line 7: 'following' is misspelled.**

   Corrected, thanks.

6. **P. 5, line 5/6: Using '$T_{atm}$' and 'atmospheric temperature' interchangeably is confusing, as is using '$T_{2m}$', 'air temperature', and 'near surface temperature'. This is especially true when, for example, the text describes 'near surface temperature' and refers to a figure in which that same variable is now called '$T_{2m}$'. Please change them to be consistent.**

   We now consistently use the acronyms $T_{surf}$, $T_{2m}$ and $T_{atm}$ wherever possible to avoid confusion.

7. **P. 6, Eq. 4: $s_{atm}$ is the dry static energy at $T_{atm}$, right?**

   Yes. We now state this in the main text.

8. **P. 9, Line 24: I don't quite understand the sentence 'Part of the difference....'. The responses of $T_{surf}$ and $T_{2m}$ are different with (Fig. 2) or without (Fig. 1) averaging, so I don't see how averaging could explain the different responses.**

   We agree with the reviewer and we removed this sentence.

9. **P. 9, Line 29: Fig. S2 is for annual mean, and doesn't say anything about daytime or 'lowest atmospheric layer' (that's $T_{atm}$, correct?). Should this refer to Fig. 2?**
   **P. 9, line 30: 'deforestation further increases surface temperature (Fig. 1)'. Should this refer to Fig. 2?**

   We are sorry that the figure numbers were wrong. We now refer to Fig. S4 and Fig. 2.

10. **P. 9: Fig. 2 shows the maps for DJF and JJA, but you don't explain why they look as they do (as is done for $T_{min}$ and $T_{max}$). I'm assuming that the $T_{max}$ effect ($\Delta T_{surf} > 0$, $\Delta T_{2m} < 0$) demonstrated**

*in Fig. S5b explains the areas where $T_{surf}$ and $T_{2m}$ are different signs in JJA, while the $T_{min}$ effect ($\Delta T_{surf} < 0$, $\Delta T_{2m} < 0$) explains why $T_{surf}$ and $T_of2m$ look similar in DJF, correct?*

We now added a paragraph hypothesizing why the maps for DJF and JJA may look different. Analogous to $T_{min}$, in the northern-hemisphere DJF often $T_{surf}<T_{2m}$, while analogous to $T_{max}$, in northern-hemisphere JJA often $T_{surf}>T_{2m}$. The reason why $T_{2m}$ responds similar to $T_{surf}$ in northern-hemisphere DJF but not JJA may be similar to the reason why $T_{2m}$ responds similar to $T_{surf}$ for $T_{min}$ but not for $T_{max}$.

11. *P. 11, line 19: I don't see how Fig. S6 shows what is described in that sentence. MPI-ESM-LR represents a completely different simulation than the one done by the authors with MPI-ESM, and it's hard to make out any details on these plots in any event.*

We removed the reference to Fig. S6 in this sentence.

12. *P. 9, Line 31: I am still not satisfied with the explanation of how $T_{2m}$ can cool for $T_{max}$ while $T_{surf}$ rises. The article again invokes the scheme used to interpolate $T_{2m}$ using $T_{surf}$ and $T_{atm}$ , but there is still no real explanation as to how in reality an increase in surface heating would not lead to an increase in both vertical mixing and $T_{2m}$, especially over a distance as small as 2m. Fig. 2 implies that the $\Delta T_{surf} > 0$, $\Delta T_{2m} < 0$ effect in the midlatitudes exists for JJA, but the other climate models do not show this (Fig. 3). You refer to the work of Meier et al. as having seen such an effect, but that too was only seen in a model, and they mention that observations contradict this. This is being put forth as a major reason that $T_{surf}$ and $T_{2m}$ differ, so I think a better explanation is needed.*

We now provide an explanation as to how in reality an increase in surface heating could lead to a decrease in $T_{2m}$, see our answer to comment 1. We now acknowledge that the MPI-ESM in Fig. 2 (different sign for the response of $T_{surf}$ and $T_{2m}$) are in contrast with the inter-model comparison of Fig. 3 (same sign for the response of $T_{surf}$ and $T_{2m}$ across the investigated models) (p. 12, l. 1). However, our main point still remains valid: that the local deforestation response of temperature is different for $T_{surf}$ and $T_{2m}$ in climate models, and that this is not specific to the MPI-ESM model but can (at least quantitatively) also be seen in other climate models.

13. *Figure S1: I have several questions about this figure:*
*1. The colored areas represent the potential vegetation from the Pongratz study, correct?*
*2. Your response to me (#3) seems to imply that forest world was created by simply replacing all current grasslands with forest, and deforested world was created by replacing current forest with grasslands. On page 4, line 8, however, your reference to the Pongratz study implies that a recreation of pre-industrial vegetation was done, which would be something different.*
*3. What do the dots mean? Are they centers of the grid boxes, or do they have something to do with the 3/4 deforestation pattern?*
*4. In the Figure S1 caption: "The 100% forest is replaced by 100% grasslands...". This is only true for 3 of every 4 grid boxes, correct?*

1. and 2.: Yes, you understood us right: the vegetated fraction of a grid cell is taken from Pongratz et al. (2008). The only modification of the Pongratz map is that the areas of all non-forest (grass, shrub, tundra) –for the forest world– or all non-grass types –for the grass world– was filled with forest or grass, respectively. The relative share of the different forest or grass types based on the share of different forest types or different grass types relative to each other as given in (Pongratz et al., 2008). We realize the sentence from page 4 is confusing. Therefore we reformulated this sentence to

make clear that the forest and grass world maps were derived directly from Pongratz et al. (2008) without any 'reconstruction' as insinuated by the unlucky wording in that sentence.

3.: We now state in the figure caption that the gray grid boxes are the ones where forest from the forest world map is left unchanged (1 of 4 grid cells).

4.: Correct. We now clarified the figure caption.

**REFERENCES**

[revised manuscript text omitted]

0.0    0.2    0.4    0.6    0.8    1.0
Vegetation cover fracton

Figure S1:  Map  showing the  fraction to which grid cells can potentially be covered with vegetation (after Pongratz et al. (2008)).  From this, the map of the 'forest world' underlying our simulations is  constructed by  replacing in 3 out of 4 grid cells the non-forest vegetated part  by forest; in the  4th grid  cell (gray) the  vegetation  distribution of the  forest world is left unchanged.
Right:  This map underlies the  zonal averages shown in  Fig. S2. The yellow color indicates grid cells where in the left map the vegetation cover fraction is larger than 50%  only these grid cells are used for  averaging.  all other grid cells, vegetation cover is less than 50%.

[Figure]

Figure S2: Zonal averages of changes in annual mean temperature for deforestation in 3 of 4 grid boxes in the MPI-ESM. Values are averaged over areas in which more than 50% of a grid box is covered by vegetation. The deforestation grid boxes and the areas that are used for the zonal averages are shown in Fig. S1.

[Figure]

Figure S3: Local effects of deforestation in the MPI-ESM, annual mean 2m-air temperature divided by surface temperature. Values below zero indicate areas where the responses of the two variables differ in sign. Values above one indicate areas where 2m-air temperature responds stronger than surface temperature.

[Figure]

Figure S4: $T_{surf}$-$T_{atm}$ as a measure of near-surface atmospheric instability, separately for nighttime and daytime conditions in the 'forest world' simulation. During nighttime, the surface is cooler than the lowest atmospheric layer in most regions. During daytime, the surface is warmer than the lowest atmospheric layer in most regions. In the maps, some regions exhibit a different sign than the rest of the world, possibly because the comparison here is not perfectly consistent: $T_{min}$ and $T_{max}$ at the surface may be reached earlier than in the lowest atmospheric layer. However, in the main text only differences between the forests and grasslands are considered, and a possibly different timing of $T_{min}$ and $T_{max}$ matters less for this difference.

[Figure]

Figure S5: Illustration of the local effects of deforestation in the mid-latitudes on different temperature variables in the MPI-ESM, separately for a) mean daily minimum temperature, and b) mean daily maximum temperature. The 'forest' values are taken from the forest world simulation and the 'grass' values are the 'forest' values plus the local effects of deforestation on the respective variable. Values are averaged over mid-latitude areas (40-60° N) that experienced intense deforestation ($\geq 15\%$) since 1860.

[Figure]

Figure S6: Maps for the **annual** means from which the averages in Fig. 3 and Table S1 were obtained. Local deforestation response of near-surface air temperature (left), surface temperature (middle) and the ratio between the two (right).

[Figure]

Figure S7: Maps for the **northern-hemispheric winter (DJF)** means from which the averages in Fig. 3 and Table S1 were obtained. Local deforestation response of near-surface air temperature (left), surface temperature (middle) and the ratio between the two (right).

[Figure]

Figure S8: Maps for the **northern-hemispheric summer (JJA)** means from which the averages in Fig. 3 and Table S1 were obtained. Local deforestation response of near-surface air temperature (left), surface temperature (middle) and the ratio between the two (right).

[Figure]

Figure S9: Temperature response to the local effects of deforestation, separately for minimum and maximum temperature during boreal winter ($T_{min}$ DJF and $T_{max}$ DJF) and summer ($T_{min}$ JJA and $T_{max}$ JJA).

**References**

Pongratz, J., Reick, C. H., Raddatz, T., and Claussen, M. (2008). A reconstruction of global agricultural areas and land cover for the last millennium. *Global Biogeochemical Cycles*, 22:1–16.